# A hydrogeological conceptual model of aquifers in catchments headed by temperate glaciers

Aude Vincent[1, 2], Clémence Daigre[2], Ophélie Fischer[2], Guðfinna Aðalgeirsdóttir[1], Sophie Violette[2, 3], Jane Hart[4], Snævarr Guðmundsson[5], Finnur Pálsson[1]

[1]Institute of Earth Sciences, University of Iceland, Askja, Sturlugata 7, IS-101 Reykjavík, Iceland

[2]Geology Laboratory, École Normale Supérieure - PSL & CNRS, UMR.8538, 24 rue Lhomond, 75231, Paris Cedex, France

[3]UFR.918, Sorbonne University, 4, Place Jussieu, 75252, Paris Cedex, France

[4]Geography and Environmental Science, University of Southampton, Southampton SO17 1BJ, UK

[5]South East Iceland Nature Research Center, Nýheimar, Litlubrú 2, IS-780 Höfn í Hornafirði, Iceland

*Correspondence to*: Aude Vincent (aude.vincent@normalesup.org)

**Abstract.** For reliable forecasting of the evolution of critical water resources and potential floods and landslides hazards and their response to climate changes, it is necessary to improve the understanding and quantification of unknown aquifer systems in glacierized catchments. We focus on four south-eastern outlet glaciers of the main Icelandic ice cap, Vatnajökull. A multidisciplinary approach is carried out, including the acquisition of new in-situ data to characterize aquifers and their groundwater dynamics. Moreover, the recharge to aquifers from glacial melt and effective rainfall is estimated. From detailed analysis of all available data and determination of the dynamic characteristics of the aquifers a hydrogeological conceptual model of glacierized catchment is constructed: (i) Two distinct aquifers, their hydraulic conductivities and their hydrodynamic response to climate forcing are identified; (ii) A comprehensive water balance for the whole catchment has been obtained; (iii) The subglacial recharge to the aquifers is 4 times higher than in the proglacial area; And (v) The importance of the impact of the glacial melt recharge on the groundwater system is demonstrated. Thus, we highlight the major role that the groundwater component has in the hydrodynamic functioning of glacierized catchments.

**Keywords.** groundwater; glacier; recharge rate; hydraulic conductivity; data acquisition; climate change; Iceland

## 1. Introduction

The research addressing the response of glaciers to climate change is well-developed, looking not only at changes in glacier mass balance (e.g.: Aðalgeirsdóttir et al., 2020; Björnsson et al., 2013; Gardner et al., 2013; Zemp et al. 2019), but also at the associated effects on basal and downstream hydrology (e.g.: Li et al., 2015, Immerzeel et al., 2012). Subglacial aquifer systems, which are inaccessible to direct measurement due to the covering glacier thickness, remain the great unknowns in the general hydrological dynamics of glaciers, their catchment areas, and their outlets. However, aftermath changes to the groundwater component are rarely considered (Vincent et al., 2019), though evolving groundwater recharge, discharge, and storage in glacierized catchments are required to forecast the future changes in water resources and water-related hazards (landslides, floods, and water shortage) as they respond to climate change. The scope of the studies considering the groundwater component is limited as they only concern the aquifers of the till/sand formations, thus neglecting the deeper underlying aquifers. Those studies in similar glacierized catchments show high recharge to aquifers by glacial meltwater (Somers et al., 2016; Mackay et al., 2020; Sigurðsson, 1990), and a strong connection between surface water (rivers or lakes) and groundwater (e.g.: Hood et al., 2006; Dzikowski and Jobard, 2012; Somers et al., 2016; Dochartaigh et al., 2019; see Vincent et al., 2019 for other references). Nevertheless, such studies are few and data relate only to shallow and unconfined aquifers (Favier et al., 2008; Dochartaigh et al., 2019; Mackay et al., 2020). What happens, if the hydrogeological system is multi-layered? Does the surface meltwater recharge the subglacial aquifer(s) or not? How much of the surface meltwater contributes to recharging the aquifer(s)?

To answer these questions, this study focuses on four outlet glaciers (Fláajökull, Heinabergsjökull, Skálafellsjökull, and Breiðamerkurjökull) at the southeast margin of Vatnajökull, Iceland's largest ice cap. The catchment areas of these outlets cover 1300 km$^2$ (fig. 1). They are located in maritime sub-polar climate and therefore experience and record the impact of climate change before those in the polar regions and thus play the role of climate change sentinel. A better understanding of how these glaciers respond to climate change contributes to a better understanding of how other glaciers around the world will respond. These glaciers are temperate and alpine, i.e.: valley glaciers with their base at melting temperature (Cuffey and Paterson, 2010). They have been retreating since the mid-1990s due to climate change (e.g.: Aðalgeirsdóttir et al., 2020; Björnsson et al., 2013). The geology in this area is mainly composed of basalt, topped in some locations by till and glacio-fluvial deposits (Jóhannesson and Sæmundsson, 1998). These formations extend into the Atlantic Ocean to form the marine abrasion platform (Van Vliet-Lanöe, 2020).

Our goal is to understand and characterize the whole hydrogeological system, the geometry, and the hydraulic parameters of the aquifers, and to quantify the recharge both in the subglacial and proglacial areas. To this end: (i) New data has been collected since May 2021 on groundwater level, temperature, and electro-conductivity (EC) in an observation network of 18 boreholes, including 4 new boreholes drilled for this project; (ii) All existing and newly acquired data is analyzed to obtain the extent and thickness of the geological formations (from geological and drilling data), the recharge rates (from existing data: glacier mass balance and weather data) and hydraulic parameters (from existing grain size data, new slug tests and new groundwater level data); (iii) All these data, of varied and complementary origins, are brought together and allow us to build a conceptual model for the hydrogeological functioning of catchments in glacial environments, and to demonstrate the importance of the groundwater component in these glacial catchments.

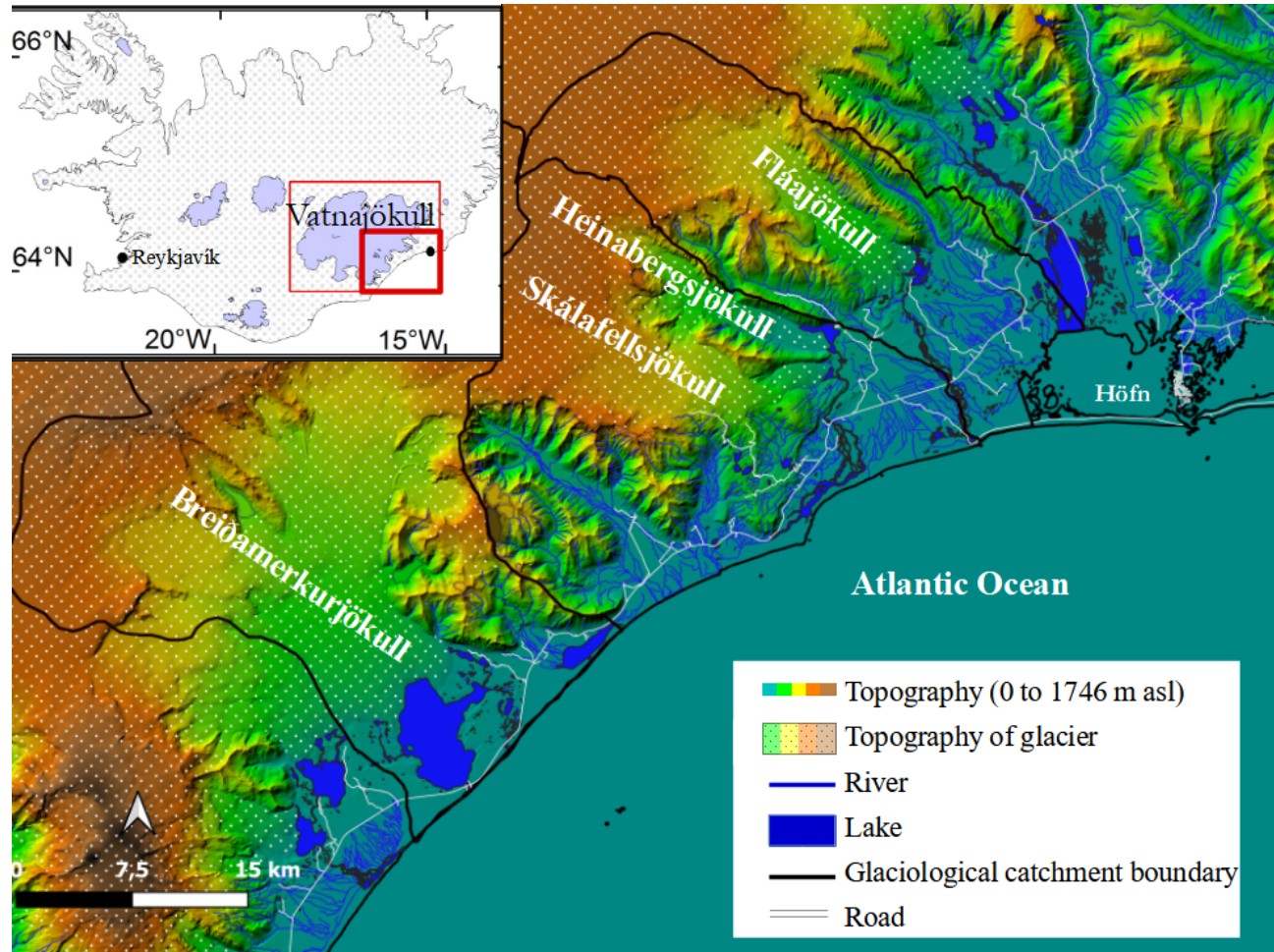

**Figure 1: The study area showing the topography of the proglacial areas and the glaciers (IslandsDEMv1, Landmælingar Íslands), glaciological catchments boundaries of the four studied outlet glaciers, river network and lakes (contours Landmælingar Íslands ISN2016). Insert map on the top left corner shows the location of the study area in Iceland, compared to Reykjavík and Höfn.**

## 2. The study area

### 2.1. Climate context

The climate in Iceland is a maritime, subpolar climate, strongly moderated by the Gulf Stream influence (Irminger current) to the south (Björnsson, 2017; Van Vliet-Lanoë et al., 2021). Snow is abundant during winter, especially over 400 m above sea level (a.s.l.) (Van Vliet-Lanoë et al., 2021), and precipitation increases with elevation (Crochet et al., 2007).

Meteorological data used in this study are provided by the Icelandic Meteorological Office (IMO). In particular air temperature (T) and precipitation (P) have been recorded at four weather stations close to Höfn í Hornafirði since June 1965 (fig. 2). The four stations are close to each other (fig. 3) but there is never the same period recorded, except for T between 2007 and 2018 (stations 705 and 5544, fig. 2), with very similar data (correlation coefficient of 0.998). Therefore, we decided to merge the time series of the four stations into one record. The combined daily temperature and precipitation records are from June 1965 to September 2022, with only four months missing (September to December 2006).

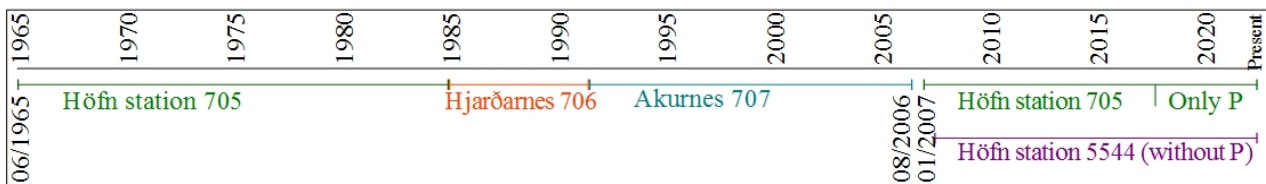

**Figure 2: Time periods of the T and P records of the 4 Höfn weather stations: Höfn 705 (+4 m a.s.l.), Höfn 5544 (+5 m a.s.l.), Hjarðarnes 706 (+9 m a.s.l.) and Akurnes 707 (+17 m a.s.l.). The location of the four stations is shown with red plus symbols in fig. 3.**

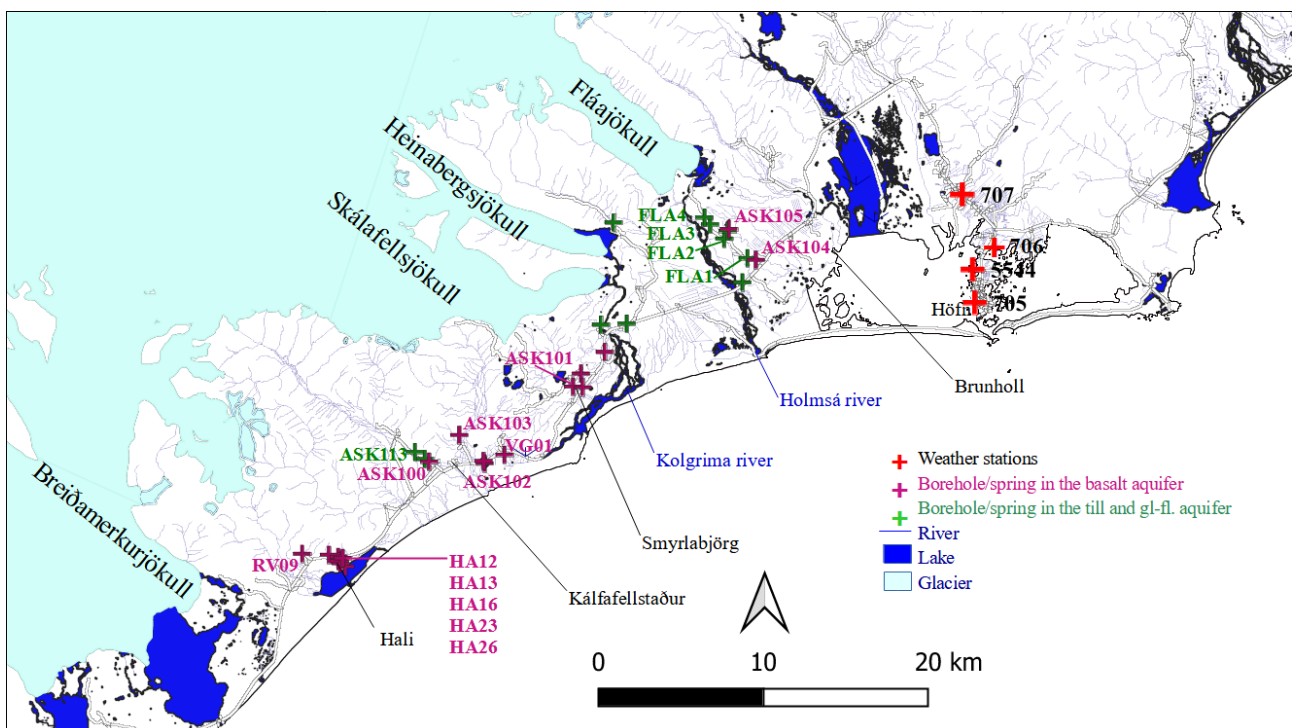

**Figure 3: Map of the groundwater observation network, Purple plus symbols: boreholes and springs in the basalt aquifer, Green plus symbols: boreholes and springs in the till and glacio-fluvial deposits aquifer (gl. Fl.), with location of Höfn weather stations (red plus symbols): Höfn 705, Höfn 5544, Hjarðarnes 706 and Akurnes 707. Contours from Landmælingar Íslands ISN2016.**

The meteorological data between 1966 to 2021 have the following characteristics: The annual average temperature has increased from 3.9 to 5.4 °C (+1.5 °C, fig. 4), and the annual total precipitation has increased from 1200 to 1630 mm (+430 mm, fig. 4). For comparison, the global mean temperature has increased by +1.08 ± 0.13 °C between 1850–1900 and 2021 (WMO, 2021). For the studied area, the annual average temperature and total precipitation in 2021 (5 °C and 1327 mm) lie in the 40 % warmest and the 40 % driest years (fig. 4). The mean monthly temperatures are distributed on a quite narrow range (fig. 5a), and nearly always stay positive, which is typical for a subpolar maritime climate. All months receive significant precipitation, but the variation between years is high (fig. 5b).

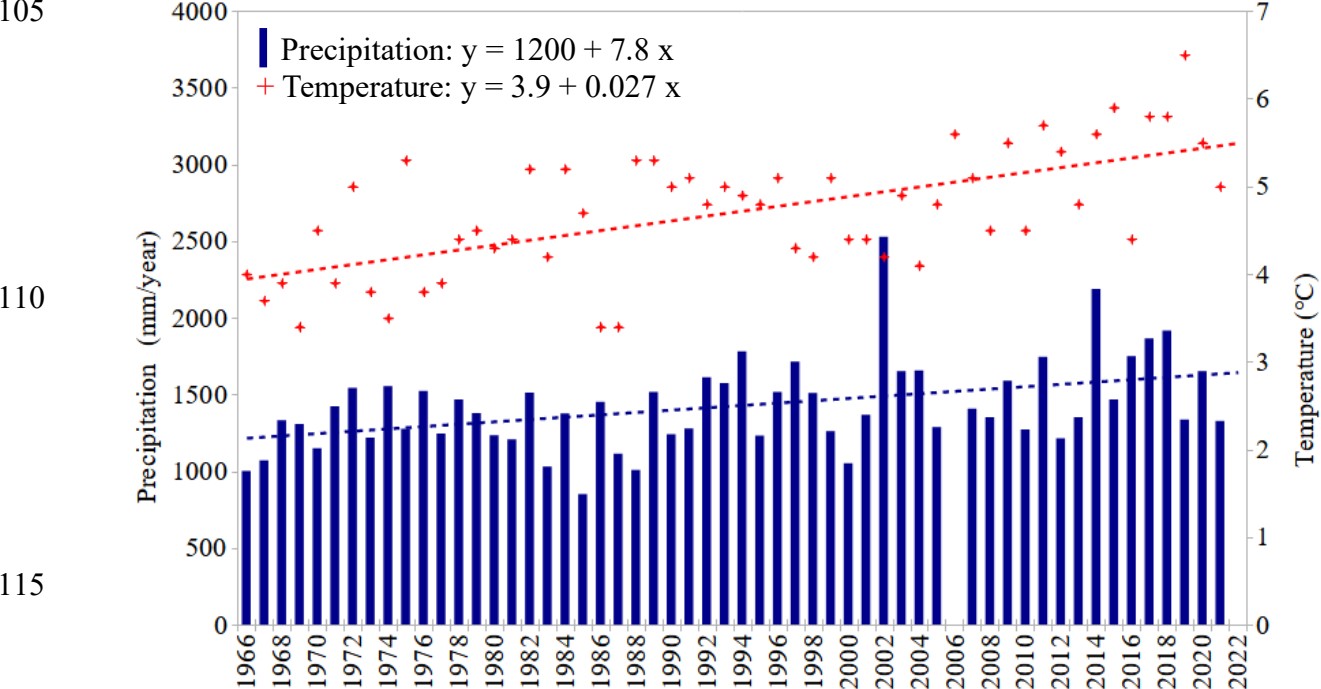

**Figure 4: Temperature and precipitation records in Höfn from 1966 to 2021 (combined record of 4 stations): Annual average temperature (red plus symbols, mean: 4.7°C) and Annual total precipitation (blue histograms, mean: 1420 mm), with linear trends (dot lines: red for temperatures and blue for precipitations).**

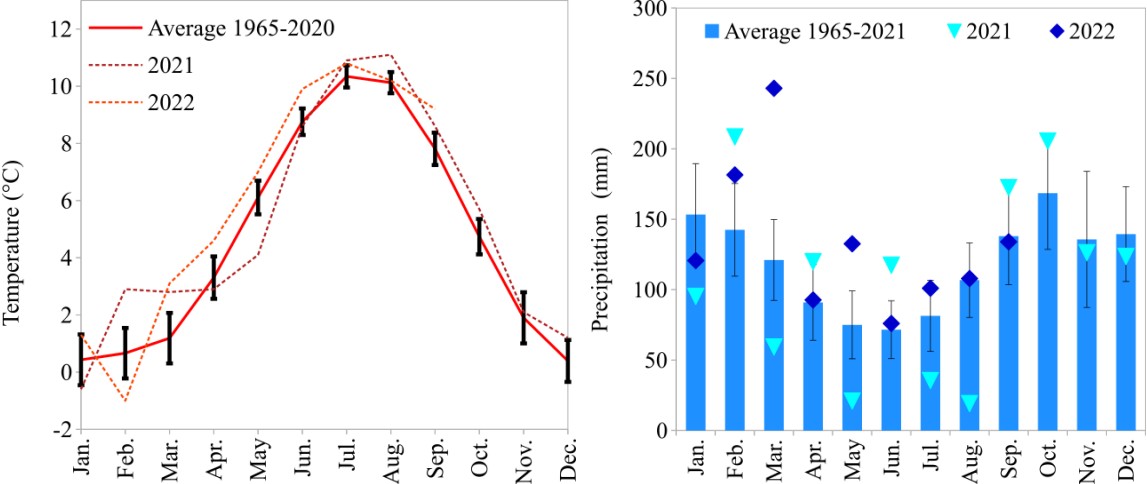

**Figure 5: The mean monthly temperature and precipitation measured in Höfn from 1966 to 2020 (combined record of 4 stations), with monthly standard deviation, and 2021 and 2022 values: a) Temperatures (monthly standard deviation from 0.7 °C to 1.8 °C); b) Precipitation (monthly standard deviation from 41 to 98 mm).**

## 2.2. Glaciers context

The four outlet glaciers (Fláajökull, Heinabergsjökull, Skálafellsjökull, and Breiðamerkurjökull) are temperate and warm based. They have undergone a complex evolution since their last maximum extent at the end of the 19th century (Hannesdóttir et al., 2015). Their recession since the mid-1990s (Björnsson et al., 2013) is linked to anthropogenic climate change (e.g.: Aðalgeirsdóttir et al., 2020). The rate of mass loss has slowed down since 2010 (Noël et al., 2022), demonstrating the high sensitivity of glaciers to climate and ocean temperature. The retreat rate is predicted to increase

again in the coming decades (Noël et al., 2022). Breiðamerkurjökull has a particularly fast retreat rate because of its unique situation, with its proglacial lake in direct connection with the ocean, whose salinity and variable temperature are enhancing the melting rate (Guðmundsson et al., 2020). Mass balance records for the glaciers (here only summer mass balance) are extrapolated from stake measurements (glaciological method, Cogley et al., 2011) on Vatnajökull ice cap (Björnsson et al., 1998; Björnsson et al., 2013; Pálsson et al., 2022) in the period 1992/93–2021.

The topographic map used for the proglacial area and the surface topography of the glaciers is IslandsDEMv1 (fig. 1), a seamless and bias-corrected mosaic from ArcticDEM (Porter et al., 2018) and lidar (Jóhannesson et al., 2013) from the National Land Survey of Iceland (IslandsDEMv1, Landmælingar Íslands), with a 2×2 m resolution and a vertical accuracy better than 0.5 m. The subglacial topography of the 4 outlet glaciers is interpolated from radio-echo sounding measurements at a resolution of 200×200 m (Björnsson and Pálsson, 2020).

## 2.3. Geological context

These glaciers lie on volcanic rocks (Einarsson, 1994) and volcano-detritic deposits that result from the interaction of the plate boundary, the Mid-Atlantic Ridge, and the Iceland Mantle Plume, that have formed the Iceland Plateau (Martin et al., 2011; Sæmundsson, 1979). Four main groups of geological formations exist (Einarsson, 1994), of which 2 are represented in the study area: Tertiary Basalt formations and sediments. The basalt encountered in the study area was formed during the Miocene and Pliocene (-13 Ma in the east of Iceland to -3.3 Ma for the west margin of Vatnajökull (Torfason, 1979)). This pile, of a total thickness up to 12 km (Torfason, 1979), and gently inclined towards the north-west (Torfason, 1979), is mainly composed of basaltic lava flows, with numerous acidic intrusions and intercalated sediments which have been largely eroded (Jóhannesson and Sæmundsson, 1998). The Vatnajökull outlet glaciers have carved out steep-sided valleys in the basaltic plateau (Hannesdóttir et al., 2010). In the study area, the basaltic plateau reaches a maximum elevation of 1746 m a.s.l. (glacier thickness included). Till is the sediment formed, transported, and deposited by glacier movements (Goldthwait, 1971), with little or no sorting by water (Dreimanis, 1983). In the study area, the proglacial areas between the glaciers and the coast are sandur (Hannesdóttir et al., 2010), with geomorphological features encompassing till and glacio-fluvial deposits. The type of soil developed are Vitric Andosol, Leptosol, and Andosols (Arnalds, 1999; Arnalds, 2015). They are partly cultivated (Hannesdóttir et al., 2010). The periglacial landforms are numerous and could induce variations in hydraulic parameters. South of Breiðamerkurjökull in particular there is a large sandur with many drumlins and eskers.

## 3. Methodology

For sustainable water management and a precise assessment of the impact of climate change on glacierized catchments a good understanding of the water cycle and thus better quantification of the water fluxes between the glacier surface and subglacial components is crucial. To accomplish those critical needs, a multidisciplinary approach based on available and new data collection is developed. The already available data provides information on meteorological conditions, glacier evolution, and geology, but does not allow characterization of the aquifers, their hydrodynamics, or their dependence on glacier melt. To better characterize the groundwater component new data acquisitions were done before analyzing the complete data set. The data analysis concerns the estimate of the subglacial melt discharge, the effective rainfall rate on the proglacial area, and the determination of the hydrodynamic properties of aquifers.

### 3.1. New Data
### 3.1.1. Aquifers geometry

The geological map of Iceland at a scale of 1/500 000 was published by the Icelandic Institute of Natural History (Jóhannesson and Sæmundsson, 1998), and is to date the only published geological map covering the study area. The drilling logs of 66 boreholes, drilled for different purposes in the past, were collected through the ISOR database (maps.is/os) or directly at the archives of the drilling company Ræktunarsamband Flóa og Skeiða (RFS). Twenty-four of these logs provide validation of the geological map and data on the thickness of the geological formations. Field data have been gathered from May 2021 to September 2022. Using these data, we have updated and simplified the geological map (Jóhannesson and Sæmundsson, 1998) to present the two main geological formations of potential aquifers: i) The detrital formation results of combined subglacial and proglacial tills, the moraines and glacio-fluvial deposits; and ii) The basalt bedrock (including the acidic intrusions). The extent of the outcrops of the till and glacio-fluvial deposits formation and of the basalt formation as indicated on the geological map have been validated in the field wherever possible (13 outcrops) and slightly corrected. The resulting map is presented in fig. 6. The thicknesses of the geological formations were estimated by combining the thicknesses extracted from the existing geological logs and the literature (Bogadóttir et al., 1987; Boulton et al., 1982; Evans et al., 2000).

### 3.1.1. Aquifers dynamics and properties

The observation network (table 1 and fig. 3) consists of 14 abandoned boreholes and 4 new ones that were drilled for this project in front of Fláajökull, on a line from the glacier terminus toward the coast (see fig. 3, FLA4, FLA3, FLA2, and FLA1). The network thus consists of 13 boreholes in the basalt formation and 5 in the till and glacio-fluvial deposits formation. In addition, we monitored 11 depression springs, 4 from the basalt aquifer and 7 from the till and glacio-fluvial deposits aquifer. Monthly manual measurements were made from March or May 2020 to September 2022 of the groundwater level, water temperature, and water EC with a manual piezometric probe (Solinst TLC Meter 107). The springs were controlled visually, and their temperature and EC were monitored with the TLC Meter. From September 2021 to September 2022, we monitored 4 boreholes in the till and glacio-fluvial deposits and 7 boreholes in the basalt using automatic pressure and temperature probes, at an hourly time step (6 TD-Diver type DI802, 3 TD Micro-Diver type DI501) and 2 probes measuring, in addition, electro-conductivity (2 CTD-Diver type DI217). To correct for the atmospheric pressure, we used 3 Baro Diver type DI800 and 1 Baro Micro-Diver DI500. A participatory approach involving Glacier Adventure, a tourism and educational company based in Hali, allows the monthly monitoring of 3 of the boreholes in the basalt aquifer all year round with a Hydrotechnik Water level meter. The locations of all boreholes and springs were measured using a differential GPS instrument in September 2021 (elevations shown in table 1).

Finally, slug tests were carried out in July and September 2022 in the 5 boreholes in the till and glacio-fluvial deposits and in 6 boreholes in the basalt formation, using slugs (diameters 10 cm and 5 cm) and an automatic pressure probe. One to three repetitions of the test were conducted in each borehole, with data recorded every half second.

| Borehole/Spring name | Elevation (m a.s.l.) | Depth (m) | Diameter (m) | Borehole rim height (m) | Year of drilling | Aquifer type | Confined /Unconfined |
|---|---|---|---|---|---|---|---|
| FLA1 | 8.74 ± 0.04 | 5.6 | 0.13 | 0.15 | 2021 | Till and gl-fl | Unconfined |
| FLA2 | 16.49 ± 0.27 | 6 | 0.13 | 0.19 | 2021 | Till and gl-fl | Unconfined |
| FLA3 | 24.11 ± 0.25 | 5.7 | 0.13 | 0.36 | 2021 | Till and gl-fl | Unconfined |
| FLA4 | 26.02 ± 0.13 | 5.8 | 0.13 | 0.19 | 2021 | Till and gl-fl | Unconfined |
| ASK113 | 36.53 ± 0.04 | 31.5 | 0.16 | 0.37 | 2010 | Till and gl-fl | Unconfined |
| Spring 2 Heinabergsdalur | 73.21 ± 0.45 | _ | _ | _ | _ | Till and gl-fl | Unconfined |
| R1 near ASK105 | 19.3 ± 0.5 | _ | _ | _ | _ | Till and gl-fl | Unconfined |
| R2 between ASK100 & ASK113 | 32.5 ± 0.5 | _ | _ | _ | _ | Till and gl-fl | Unconfined |
| R5 west of Kolgríma | 13.1 ± 0.5 | _ | _ | _ | _ | Till and gl-fl | Unconfined |
| R6 west of Hólmsá | 8.5 ± 0.5 | _ | _ | _ | _ | Till and gl-fl | Unconfined |
| R9 near Brunholl | 2.7 ± 0.5 | _ | _ | _ | _ | Till and gl-fl | Unconfined |
| ASK100 | 26.39 ± 0.08 | 56 | 0.16 | | 2010 | Basalt | Confined - Artesian |
| ASK101 | 37.77 ± 0.08 | 100 | 0.16 | 0.39 | 2010/2015 | Basalt | Confined |
| ASK102 | 15.50 ± 0.04 | 49.6 | 0.16 | 0.24 | 2010 | Basalt | Unconfined |
| ASK103 | 24.28 ± 0.09 | 55 | 0.16 | 0.31 | 2010 | Basalt | Confined |
| ASK104 | 13.59 ± 0.09 | 50 | 0.16 | 0.34 | 2010 | Basalt | Confined |
| ASK105 | 20.12 ± 0.21 | 50 | 0.16 | 0.42 | 2010 | Basalt | Unconfined |
| HA12 | 54.23 ± 0.06 | 72 | 0.16 | 0.26 | 2001 | Basalt | Confined |
| HA13 | 60.80 ± 0.07 | 49 | 0.16 | 0.43 | 2002 | Basalt | Unconfined |
| HA16 | 3.14 ± 0.06 | 54 | 0.16 | 0.12 | 2002 | Basalt | Confined |
| HA23 | 14.15 ± 0.05 | 60.5 | | 0.27 | | Basalt | Confined |
| HA26 | 26.20 ± 0.05 | 70 | 0.16 | 0.29 | 2016 | Basalt | Confined |
| RV09 | 47.35 ± 0.04 | 60 | 0.18 | 0.44 | 2016 | Basalt | Confined |
| VG01 | 11.39 ± 0.3 | 60 | | 0.34 | 2002 | Basalt | Unconfined |
| R3 Smyrlabjörg | 19.8 ± 0.5 | _ | _ | _ | _ | Basalt | Unconfined |
| Spring 1 Smyrlabjörg | 15.7 ± 0.5 | _ | _ | _ | _ | Basalt | Unconfined |
| R4 near ASK102 | 6.9 ± 0.5 | _ | _ | _ | _ | Basalt | Unconfined |
| R7 Smyrlabjörg – Kolgríma | 10.1 ± 0.5 | _ | _ | _ | _ | Basalt | Unconfined |

**Table 1: List of the groundwater observation network. For each borehole, their elevation, depth, diameter, borehole rim height, year of drilling, aquifer type, and the confined or unconfined status of the aquifer are listed. gl-fl= glacio-fluvial deposits. For the locations of the listed rivers and places see fig. 3.**

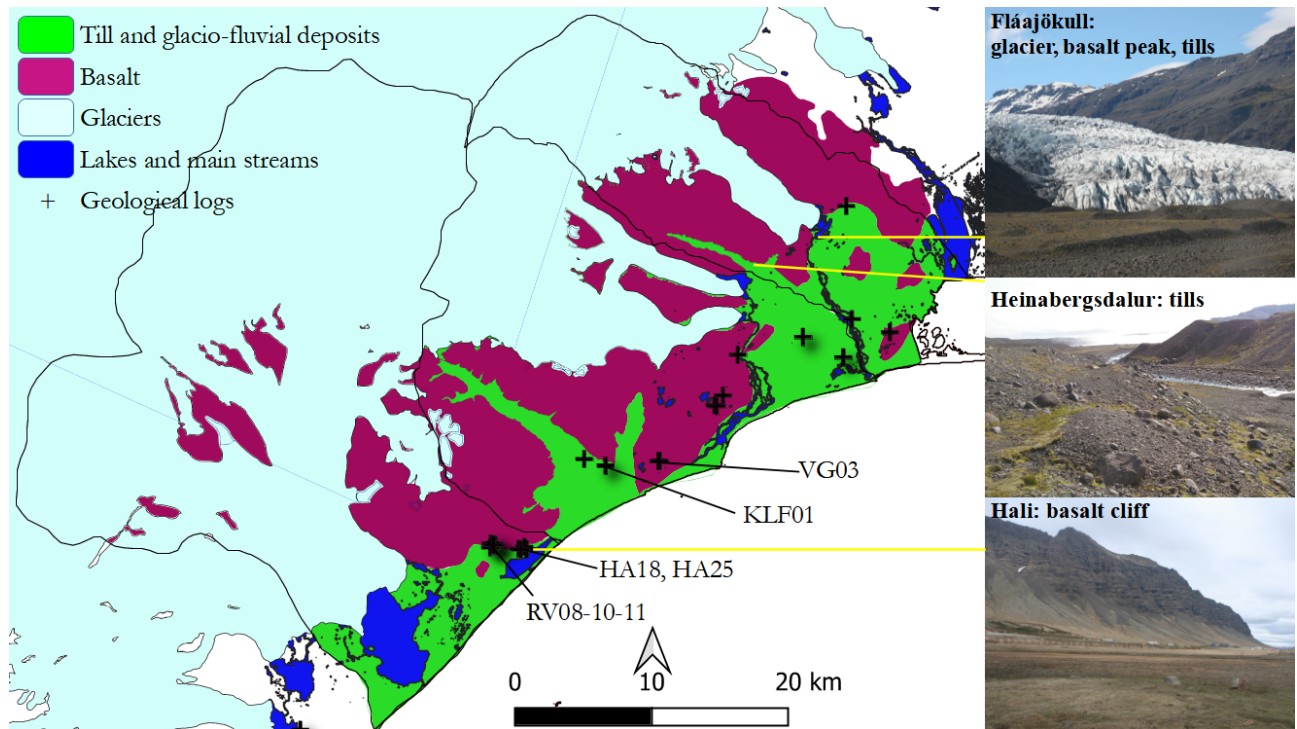

**Figure 6: The resulting geological map of the study area, with locations of the geological logs giving information on the thickness of geological formations. The basalt formation is represented in purple and the till and glacio-fluvial deposits in green. In light blue the glacier, and dark blue the lakes and main rivers. Pictures on the right side offer some typical views of the geological system: Outlet glacier (Fláajökull), tills in Heinabergsdalur, and a basalt cliff at Hali. Geological contours based on Jóhannesson and Sæmundsson 1998; other contours from Landmælingar Íslands ISN2016.**

### 3.2. Data Analysis

### 3.2.1. Glacier melt and effective rainfall

Estimates of glacier melt and effective rainfall were used to deduce the amount of available water for subglacial flow and groundwater recharge in the subglacial area. To estimate the glacier melt, we used two datasets: (i) The estimated summer mass balance based on direct glaciological methods from 2010 to 2021 (Björnsson et al., 1998; Björnsson et al., 2013; Pálsson et al., 2022); and (ii) Melt calculated with the offline HIRHAM model forced with output from the Regional Climate Model HARMONIE-AROME reanalysis-forced simulations from 1980 to 2016 by Schmidt et al. (2020) that was used to force the Parallel Ice Sheet Model (PISM) (Bueler and Brown, 2009). HARMONIE-AROME is a non-hydrostatic, convection-permitting model (Bengtsson et al., 2017), that used the reanalysis of the Icelandic Meteorological Office for Iceland (ICRA) (Nawri et al., 2017) as boundary conditions, from 1 September 1979 until 31 December 2017, at a horizontal resolution of 0.025° × 0.025°, corresponding to ~2.5 km. In the rest of the paper, these simulations by Schmidt et al., (2020) are called HH-ICRA. The non-surface mass balance is added to the HH-ICRA outputs: 0.23 mm day$^{-1}$ of dissipation for all the area, and an additional 0.15 mm day$^{-1}$ from lake calving for Breiðamerkurjökull and Heinabergsjökull (Aðalgeirsdóttir et al., 2020; Jóhannesson et al., 2020).

The quantification of the effective rainfall must first estimate the potential evapotranspiration from available climatic variables, then use the water balance method. The potential evapotranspiration (PET) was calculated using the Thornthwaite method (Thornthwaite, 1948) for evolution since 1966 and with the Penman method (Monteith, 1965; Penman, 1948, with CropWat 8.0) and for monthly values between 1990 and 2022, using available weather station data

from Höfn weather stations and parameters adapted to the latitude. We chose classic methods, as from the literature there is no specific method that proved more representative for Iceland.

Effective rainfall was then calculated, with the following equations:

if $P_i > PET_i$,                     $RET_i = PET_i$    and if $S_{i-1} = S_{max}$    $EF_i = P_i - RET_i$,

                                     otherwise       $EF_i = P_i - RET_i - (S_{max} - S_{i-1})$

if $P_i = PET_i$,                     $RET_i = PET_i$   and $EF_i = 0$

if $P_i < PET_i$ and $P_i + S_i >= PET_i$   $RET_i = PET_i$ and $EF_i = 0$

          and $P_i + S_i < PET_i$,    $RET_i = P_i + S_{i-1}$, $S_i = S_{i-1} - (PET_i - P_i)$ and $EF_i = 0$

with $_i$: month; $RET_i$: Real Evapotranspiration; $PET_i$: Potential Evapotranspiration; $P_i$: Precipitation; $S_i$: Soil water storage capacity, $EF_i$: Effective Rainfall. $S$ is initially estimated at 50mm, as an average for the Vitric Andosol, Leptosol, and Andosols that compose the area (Arnalds, 1999; Arnalds, 2015). As rainfall and snow events are not completely distinguished in the precipitation record provided by the IMO, the effective rainfall we calculate does not represent the delayed recharge to the aquifers due to the snow melt.

### 3.2.2. Hydrodynamic properties of aquifers

We calculated the hydraulic conductivities with two different methods: From grain size data for the till and glacio-fluvial deposits and slug tests for both types of aquifers.

For the grain size data method, we used $d_{10}$ from samples collected in the Skálafellsjökull area. Their representativeness is local and up to 2 m depth. We carried out the calculation using the modified Hazen formula, developed empirically (in various soil types in Northern Scotland, MacDonald et al., 2012) and applied to Virkisjökull, a glacierized valley very similar to the study area (Dochartaigh et al., 2019):

$$\log(K) = 0.79 * \log(d_{10}) + 2.1 - 0.38 * SSD$$

with $K$ the hydraulic conductivity in $m.d^{-1}$; $d_{10}$ the threshold grain size under which 10 % of the grains are represented in mm; SSD the Soil State Description value, ranked between 0 and 1, from very loose to very dense state, here SSD = 1 (Dochartaigh et al., 2019).

We interpreted the slug tests with the Bouwer and Rice solution (Bouwer and Rice, 1976) for the unconfined boreholes (only on the rising parts of the tests) and with the Hvorslev method (Hvorslev, 1951) for the confined boreholes. The results are representative for a few meters distance in radius from the borehole and for the whole depth of the screen in the borehole (table 1).

We estimated specific yield ($S_y$) for both aquifers using grain size data (graph in Robson, 1993) and the Water Table Fluctuation (WTF) method on distinct rainfall-recharge event after manual correction of the Lisse effect (Crosbie et al., 2005; Healy and Cook, 2002):

$$S_y = R / \Delta h$$

with $R$ the recharge in m, and $\Delta h$ the increase in groundwater table in m. Only recharge events exclusively due to rainfall should be considered, thus we excluded days with snow precipitation and/or snow cover, as well as periods of potential significant glacial melt recharge.

To perform these data analyses, we used several free and open software packages: the Geographic Information System QGIS (QGIS.org, 2022) to update and create maps, as well as to perform mathematical operations on maps; LibreOffice for statistical treatments and graphics drawing (Foundation, T.D., 2020); and Python (Python.org, 2022) for data analysis and drawing of graphics.

## 4. Hydrodynamic characterizations of the aquifers

We present here the new data sets on groundwater level, temperature, and electrical conductivities as well as the results of the slug tests that allow the characterization of the dynamic nature of each aquifer, the response to climate forcing, and the hydrodynamic properties.

### 4.1. Groundwater level

The till and glacio-fluvial deposits aquifer is unconfined with groundwater level that is often near or at the surface. South of Fláajökull, in the newly drilled boreholes the groundwater levels are very close to the surface (fig. 7). The amplitude of the groundwater level seasonal variations decreases with the distance to the glacier (from 1.70 m in FLA4 to 0.30 m in FLA1). Uncertainties between manual and automatic probes are smaller than 6 cm, except for FLA3 where they are of the order of 13 cm. This area is in a county that is called "mýrar", which means swamps, and many temporary swamps can be found in this area.

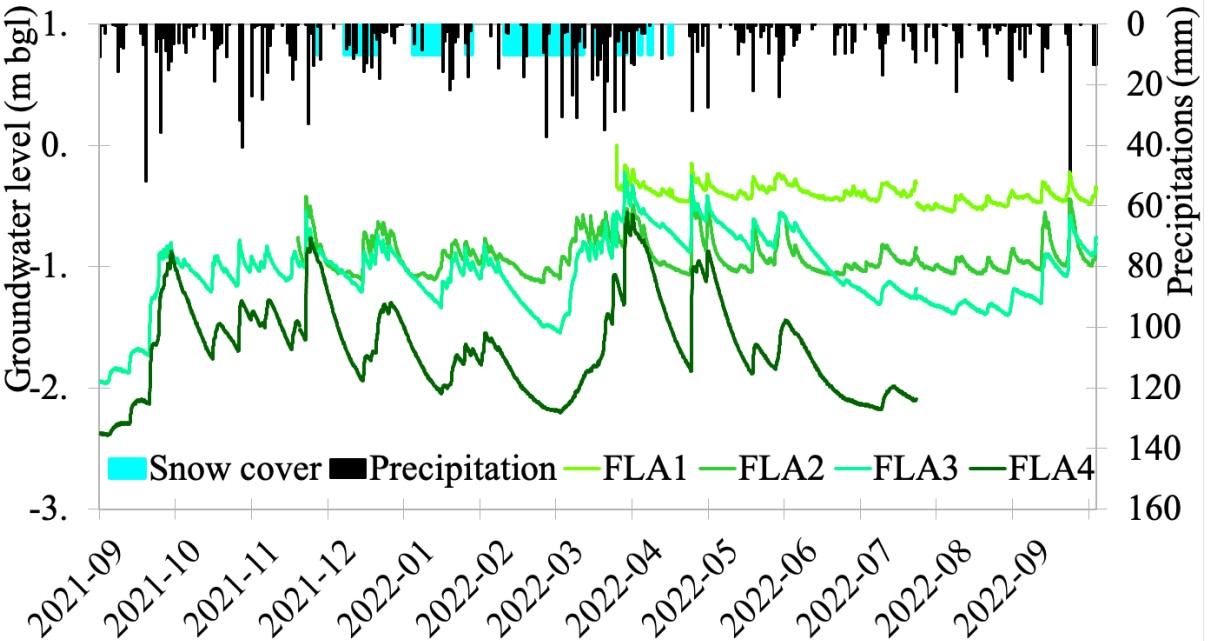

**Figure 7: Hourly evolution of the groundwater level (in m b.g.l.) in the till and glacio-fluvial deposits aquifer south of Fláajökull, from August 2021 to September 2022 (in the 4 new boreholes: FLA1, FLA2, FLA3, FLA4), daily precipitation (in mm) from Höfn weather station 705, and days with snow cover.**

Depending on the location, the basalt aquifer can be unconfined (while it is outcropping) or confined (while it is covered by till and glacio-fluvio deposits), of artesian at some locations. The groundwater level in the basalt aquifer varies from -42 m b.g.l. (below ground level) in HA13 (fig. 8, in a topography slope at the bottom of a basalt cliff) to the surface topography in some proglacial areas (ASK100 in Kálfafellsdalur valley, recurring artesian well, fig. 3), and a recurring spring in the proglacial area between Skálafellsjökull and the coastline. In the proglacial areas, the confined parts of the basalt aquifer (HA16, ASK103 and ASK104, fig. 8) show groundwater level variations with much smaller amplitudes (0.4 to 0.6 m) than in the unconfined parts (over 1 m in boreholes ASK102 and ASK105, fig. 8). In the topography slopes at the bottom of the basalt cliffs the amplitude of the variations is much larger (nearly 2 m to 4 m, boreholes HA13 and HA12, fig. 8). Uncertainties between manual and automatic probes are smaller than 6 cm for ASK013, ASK104 and HA16, and of the order of 15 cm for ASK102, ASK105, HA12 and HA13.

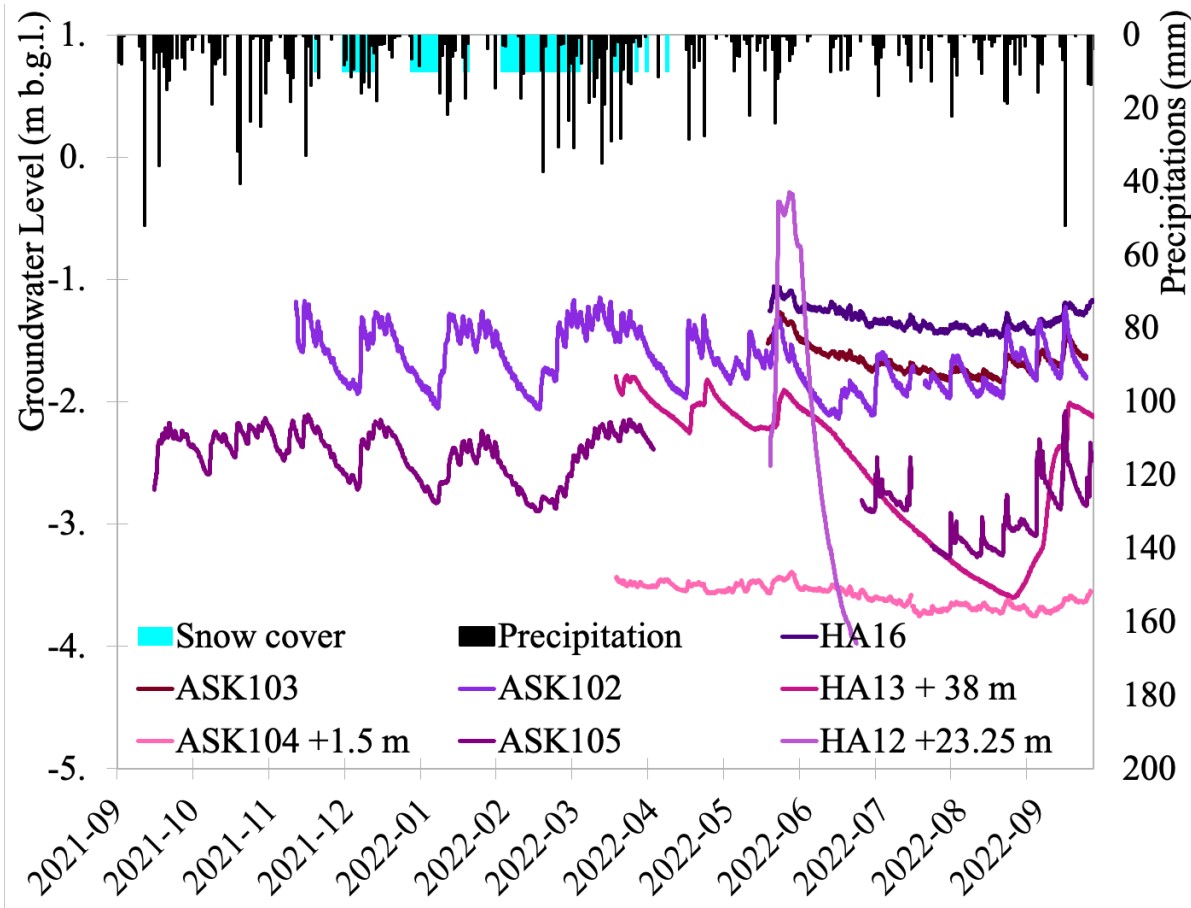

**Figure 8: Hourly evolution of the groundwater level (in m b.g.l.) in the basalt aquifer, from September 2021 to September 2022 (5 boreholes: ASK102, ASK103, ASK104, ASK105, HA16), daily precipitation (in mm) from Höfn weather station 705, and days with snow cover.**

### 4.2. Temperature

Hourly temperatures were recorded in ten boreholes: FLA2, FLA3 and FLA4 in the till and glacio-fluvial deposits aquifer, and ASK102, ASK103, ASK104, ASK105, HA12, HA13 and HA16 in the basalt aquifer (location on fig. 3), with an uncertainty smaller than 0.2°C. In the till and glacio-fluvial deposits aquifer, the temperature follows the trends of the atmospheric temperature with a lag of about six weeks: The groundwater temperature decreases from September to March and increases from April to September, covering a range of 1–9°C (fig. 9). Several plateaus of temperature

around 5°C are visible in the FLA4 record and are interpreted in sect. 5.3. All the probes in the boreholes in the basalt aquifer at less than 10 m b.g.l. (all boreholes except HA13) display a narrow range of values, 5–9°C.

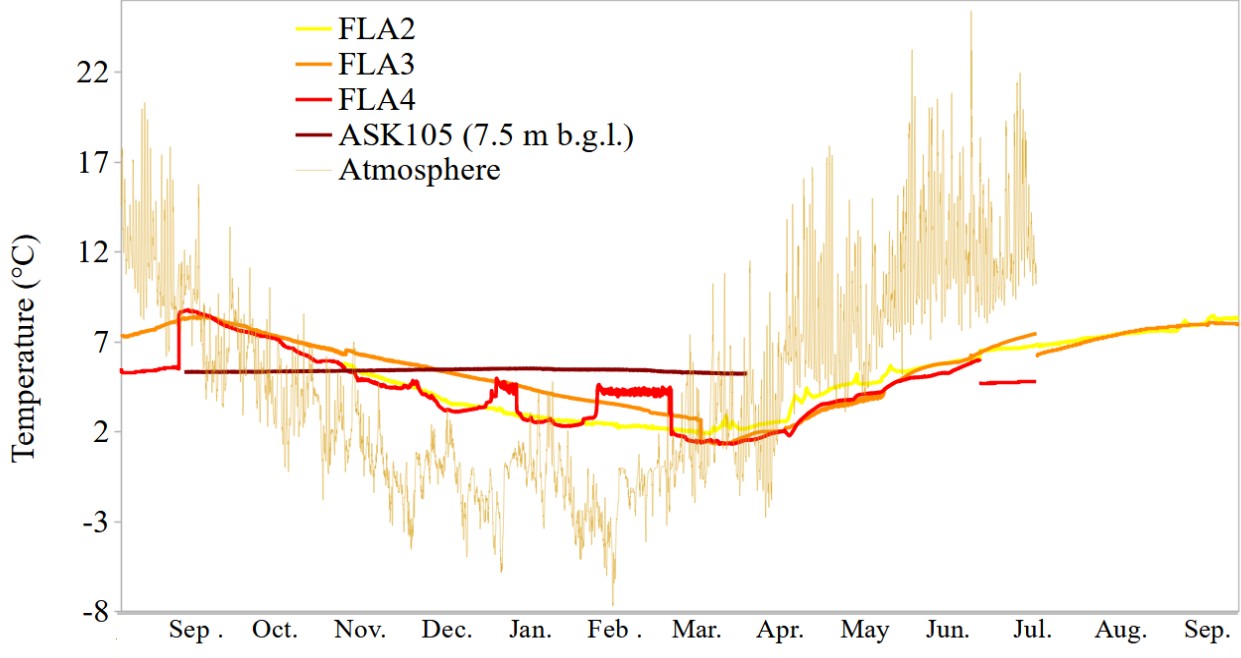

**Figure 9: Hourly temperature records from August 2021 to September 2022: in the till and glacio-fluvial deposits aquifer in 3 boreholes (FLA2, FLA3, FLA4), in the basalt aquifer in 1 borehole (ASK105 at -7.5 m b.g.l.), and in the atmosphere (recorded by a Baro Diver at the top of FLA4).**

### 4.3. Electro-conductivity

Water with an EC < 700 µS cm$^{-1}$ is considered non-saline, between 700 and 2000 µS cm$^{-1}$ slightly saline and between 2000 and 10 000 µS cm$^{-1}$ moderately saline (Rhoades et al., 1992). In table 2 the recorded EC values are shown. Values above 700 µS cm$^{-1}$ have been measured in three boreholes, by order of decreasing values: VG1, HA16, and HA23. Values measured in HA26 are just below or just over 700 µS cm$^{-1}$. In ASK104, groundwater is completely fresh until -8 m b.g.l., but below that level, EC values are significantly above 700 µS cm$^{-1}$. These boreholes are close to the coastline or to a brackish lake connected to the sea (fig. 3). The closer they are to one or the other, the higher their EC values are. EC has also been measured hourly in HA16 from May to September 2022: EC is varying from 700 to 1850 µS cm$^{-1}$ with regular cycles (period of 24 hours due to the tide and period of 3 to 4 days).

| EC (µS cm$^{-1}$) | HA16 | HA23 | HA26 | VG01 | ASK104 (at 9 m depth) |
|---|---|---|---|---|---|
| 08/2021 | 2600 | 1300 | 900 | 4400 | 1400 |
| 09/2021 | 2200 | 1100 | 700 | 5600 | _ |
| 03/2022 | 4700 | _ | _ | 6200 | 2600 |
| 05/2022 | 1800 | _ | 500 | 2600 | _ |
| 06/2022 | _ | _ | 600 | 3000 | _ |
| 07/2022 | _ | _ | 600 | 2900 | _ |
| 09/2022 | 1900 | 1200 | 600 | 3900 | 1800 |

**Table 2: Electro-conductivity (EC) values (µS cm$^{-1}$) measured in boreholes in the study area from August 2021 to September 2022; _: no measurement available. For location see fig. 3. Uncertainties +/- 100 µS cm$^{-1}$.**

### 4.4. Slug tests

Table 3 summarizes the results of the slug tests conducted in 11 boreholes. Hydraulic conductivities ($K$) of till and glacio-fluvial aquifer range from 5.8E-6 m s$^{-1}$ to 3E-5 m s$^{-1}$ while those of basalt aquifer range from 1.1E-10 to 4.9E-6 m s$^{-1}$. Thus, there is a wider heterogeneity of $K$ in the basalt aquifer, depending on their fractured degree.

| Borehole | Formation | (Un)confined | Number of tests | Analysis method | $K$ (m s$^{-1}$) | +/- | Distance to the glacier front (m) |
|---|---|---|---|---|---|---|---|
| ASK113 | Till and g-f | unconfined | 6 | Bouwer and Rice | **5.8E-6** | 1.5E-7 | 6420 |
| FLA1 | Till and g-f | unconfined | 6 | Bouwer and Rice | **5.8E-6** | 1.5E-6 | 6350 |
| FLA2 | Till and g-f | unconfined | 3 | Bouwer and Rice | **6.9E-6** | 1.0E-6 | 4520 |
| FLA3 | Till and g-f | unconfined | 3 | Bouwer and Rice | **1.7E-5** | 4.9E-6 | 3280 |
| FLA4 | Till and g-f | unconfined | 6 | Bouwer and Rice | **3.0E-5** | 1.1E-6 | 2740 |
| ASK101 | Basalt | confined | 1 | Hvorslev | **1.1E-10** | - | |
| ASK102 | Basalt | unconfined | 3 | Bouwer and Rice | **8.5E-8** | 3.8E-8 | |
| ASK103 | Basalt | confined | 4 | Hvorslev | **2.9E-7** | 1.0E-7 | |
| ASK105 | Basalt | unconfined | 5 | Bouwer and Rice | **5.4E-8** | 5.8E-8 | |
| HA16 | Basalt | confined | 2 | Hvorslev | **8.6E-7** | 8.6E-7 | |
| RV09 | Basalt | confined | 5 | Hvorslev | **4.9E-6** | 1.8E-6 | |

**Table 3: Results of the slug tests conducted in July and September 2022. For each borehole, the aquifer type, the confined or unconfined status of the aquifer, the number of tests, the method of analysis used to interpret the slug tests data, the average value of hydraulic conductivity calculated from the data, the uncertainty of the measurements, and the distance to the glacier front for the till and g-f aquifer are listed. g-f: glacio-fluvial deposits.**

## 5. Results

We will first detail the amounts of available water for surface flow and groundwater recharge in the subglacial and the proglacial areas, providing an upper limit for the recharge rates towards the aquifers, then go through the characteristics of both aquifers and finally their dynamic characteristics.

### 5.1. Recharge estimation

### 5.1.1. Estimation of subglacial water flows and spatial distribution

Combining all the data we concluded that the available water for surface flow and subglacial recharge in the subglacial area (table 4) is on average 4000 mm year$^{-1}$. During the period 2010–2016, the average available water quantity on each glacier obtained from the combination of the HH-ICRA results or the summer mass balance with the effective rainfall are very similar for Fláajökull and for Skálafells- and Heinabergsjöklar the HH-ICRA results give 10% higher values, but significantly higher for Breiðamerkurjökull, by 22% (table 4). The subglacial recharge is highly seasonally variable and peaks in July and August (fig. 10). An example of the elevation dependence of the available subglacial water is shown in table 5. As can be expected, there is much more available subglacial water at the lowest elevation of the glaciers for both data sets. The subglacial recharge thus varies both temporally and spatially, and the available data and results allow us to account for that.

| Data set | Time period | Subglacial water available (mm year⁻¹) | | |
| --- | --- | --- | --- | --- |
| | | Breiðamerkurjökull (eastern part) | Skálafells &Heinabergsjöklar | Fláajökull |
| **Summer Mass Balance** | 2010-2021 | 3600 | 4000 | 3800 |
| | 2010-2016 | 3600 | 4000 | 3800 |
| | 2021 | 3600 | 4200 | 4000 |
| **HH-ICRA results** | 2010-2016 | 4400 | 4400 | 3800 |

**Table 4: Available water for surface flow and subglacial water available (mm year⁻¹), estimated from: i) Summer mass balance measurements (field data, IES Glaciology group) and effective rainfall, ii) HH-ICRA results (Schmidt et al., 2020) and effective rainfall, for each outlet glacier considered.**

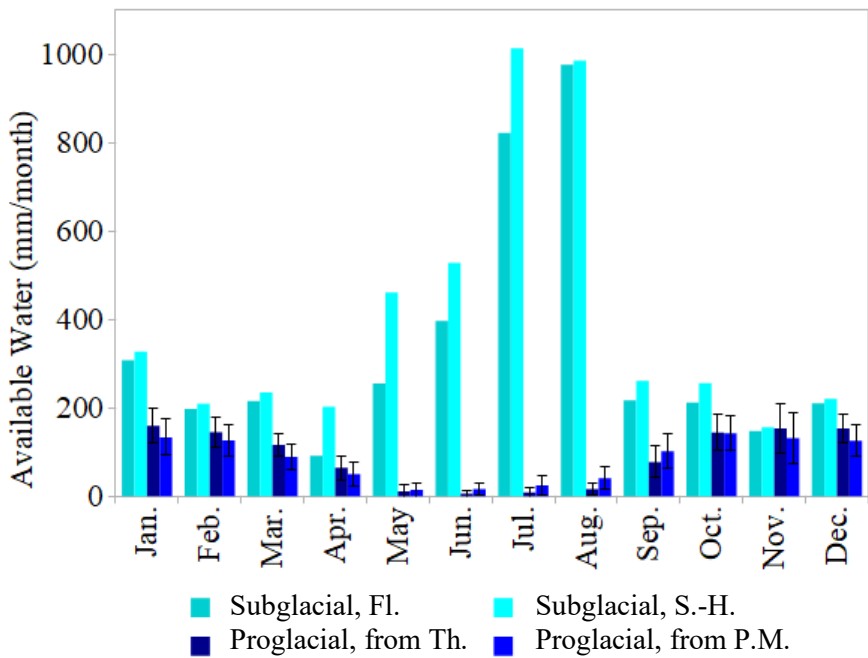

**Figure 10: Monthly mean available water in the subglacial area and in the proglacial area: (i) In the subglacial area, the values result from the estimated melt (from HH-ICRA 2012 projected with the 2021 total summer mass balance) and the effective rainfall, under Fláajökull (Fl.) and Skálafells-Heinabergsjöklar (S.-H.), (ii) In the proglacial area, the values are average over 1990-2021, calculated from monthly precipitation data (from combined Höfn weather stations) and monthly potential evapotranspiration calculated with Thornthwaite method (Th., monthly standard deviation between 30 and 192**
**mm) and with Penman-Monteith method (P.M., monthly standard deviation between 54 and 228 mm); Table 5: Elevation-dependent subglacial available water (mm year⁻¹), estimated from summer mass balances and HH-ICRA results. Example of one year (2010) for one glacier (Fláajökull) to give an insight into the variability of the subglacial available water rates with the elevation.**

### 5.1.2. Estimated effective rainfall in the proglacial area

The total effective rainfall, based on the PET calculated with Thornthwaite method, has increased by 165 mm (from 975 to 1140) since 1990, following the precipitation trend (fig. 4).

The monthly variation of the effective rainfall, based on the PET calculated from 1990 to 2021, shows that several months each year have no effective rainfall at all (PET with Thornthwaite method: 2–5 months, very rarely none; Penman-Monteith method: 1–4 months, very rarely 0, 5 or 7 months). Months with no effective rainfall are most of the

time between April and August. These months have an inter-annual average of effective rainfall < 30 mm. The winter months, October to March, all have an average effective rainfall > 100 mm, with a maximum in January and February.

| Water available in the proglacial area (mm year$^{-1}$) | From PET calculated with Thornthwaite method | | | From PET calculated with Penman-Monteith method | | |
|---|---|---|---|---|---|---|
| | 1990-2019 | 2020 | 2021 | 1990-2019 | 2020 | 2021 |
| | 1020 | 1120 | 900 | 970 | 900 | 830 |
| Uncertainties | +/- 280 | | | +/- 360 | | |

Table 6: Effective rainfall or available water (mm year$^{-1}$) in the proglacial area (between the glacier's terminus and the coastline) for both surface runoff and recharge to the aquifers.

In the proglacial area, the interannual average hydrological balance in the period 1990–2021 is the following: 1540 mm
(± 310 mm) of precipitation, 520 mm (± 80 mm) of evapotranspiration (both methods used), 1000 mm (± 300 mm) of water available for both surface runoff and for recharge to the aquifers.

### 5.2. Aquifers characteristics

### 5.2.1. Thickness and temperature of geological formations

The thickness of the subglacial till formation is estimated with Ground Penetrating Radar measurements to range from 1
to 20 m thick (average 5 m) under Skálafellsjökull (Hart et al., 2015; Hart, 2017). The proglacial tills closest to the glacier terminus, which can be a good representation of subglacial ones, have a thickness of up to 3 m in front of Skálafellsjökull (Hart, 2017), and 3.5-5 m in front of Fláajökull (Evans and Hiemstra, 2005). The thickness probably has some spatial variability and the subglacial till might not be continuous everywhere.

According to the drill logs of the 16 boreholes consulted (locations on fig. 6) the till and glacio-fluvial deposits
formation total thickness ranges from 2 to 54 m (maximum thickness in RV08) with an average of 15 m. This range is confirmed by 5 m of till and glacio-fluvial deposits in a borehole in Hali (HA25, Sigurðarson et al., 2016) and a 16 m high stratigraphic log full of sediments in a cross-section along the Kolgríma River, close to the Skálafellsjökull terminus (Evans et al., 2000). According to seismic reflection and refraction measurements south of Breiðamerkurjökull, the thickness of the till and glacio-fluvial deposits ranges between 30 m and 150 m, from the
glacier to the coastline (Bogadóttir et al., 1987; Boulton et al., 1982).

A basalt formation has generally a high contrast in hydraulic conductivity ($K$) from top to bottom, as with time and a temperature over 50°C the fractures are filled with secondary minerals, products of the chemical alteration (smectite, illite, zeolite at low temperatures and chlorite and epidote for higher grade metamorphism, e.g.: Arnórsson, 1995). Thus, we assume that below the 50°C isotherm, the hydraulic conductivity of the basalt formation becomes negligible. The
depth of the 50°C isotherm can be estimated using the drilling logs retrieved from RFS and data presented by Sigurðarson et al., (2016), who provide detailed analysis of geophysical measurements in the borehole HA25 in Hali. The temperature log of borehole HA25 indicates 51°C, at -303 m, but both the stratigraphic description and the deviations in geophysical measurements show a shallower depth of -194 m (Sigurðarson et al., 2016). From the drilling logs retrieved from RFS, we have identified temperatures higher than 50°C only in HA14: 53 °C, at -384 m (43°C, at -
186 m), suggesting a boundary at approximately -400 m. No temperature higher than 50°C is observed in HA18 (at 180 m,), RV08 (at 340 m), RV11 (at 143 m), suggesting that the 50°C isotherm is not above at least -140 m. Furthermore, weathered materials are described in VG03 (precipitate from -173 m to -200 m), suggesting a transition at -200 m, and in RV10 (dark shale at -709 m). Thus, we conclude that the current 50°C isotherm is between -300 and -400 m, and the boundary with older altered areas brought nearer to the surface by glacial erosion is around -200 m. The bottom of the

basalt aquifer can thus be considered to be between -200 m and -400 m below the surface or at the bottom of the till and glacio-fluvial deposits.

### 5.2.2. Aquifers hydraulic conductivity

The hydraulic conductivity ($K$) values of the till and glacio-fluvial deposits calculated from grain size data (fig. 11) vary from 4.5E-6 to 3E-5 m s$^{-1}$, (for average $d_{10}$ of 2 and 22 μm, respectively). This is in the range of values found in the

415 literature for Iceland: from 1E-6 m s$^{-1}$ (Iverson et al., 2017, Múlajökull till) to 7E-4 m s$^{-1}$ (Dochartaigh et al., 2012, Virkisjökull, glacial till, max. measured). The value of 3E-5 m s$^{-1}$ is very close to the 3.8E-5 m s$^{-1}$ calibrated for the Virkisjökull sandur (Mackay et al., 2020). Measurements done via slug tests yield similar results as the ones calculated from grain size data: 1.5E-5 [5.8E-6–3E-5] m s$^{-1}$. They moreover show a decreasing trend from the glacier toward the coastline (table 3), which can result from the decrease of grain size from the glacier terminus to the coast, due to fluvial

transport.

Hydraulic conductivities ($K$) values for the basalt formation calculated with the data from the slug tests are presented in table 7 and also shown in fig. 11. Outside Iceland, $K$ values for basalt have been recorded from 1E-6 to 6E-1 m s$^{-1}$ (fig. 11) and show the heterogeneous character of this type of formation. Higher values are found in fissured and young basalts, while lower ones are found in the core of lava flows, or weathered and old lavas. This range is based on the

425 following references: Columbia River Plateau (USA), on a 1 to 5 km thick Miocene plateau, with 800 bulk $K$ measurements in 577 wells on the first 500m: 6E−5 to 6E−1 m s$^{-1}$, vertical $K$ 5 times smaller (Jayne and Pollyea, 2018); La Réunion, 6E-4 to 3E-1 m s$^{-1}$ (Join, 1991), 1E-3 m s$^{-1}$ (numerical model calibration) (Violette et al., 1997); Mayotte, in situ measures in lava flow: massive <1E-6 m s$^{-1}$, scoriated: 5E-6 to 5E-4 m s$^{-1}$, fissured: 1E-6 to 5E-4 m s$^{-1}$ (Lachassagne et al., 2014). In Iceland K values for basalt vary by at least 3 orders of magnitude (fig. 11): it is less than

430 1E-10 m s-1 (Dochartaigh et al., 2012 and 2019: Virkisjökull glacier in south-eastern Iceland, through constant rate pumping tests of 3 to 6 hours), and 2E-8 to 5E-8 m s-1 (Jonsson and Hafstað, 1991) at boundary between the mountainous East Fjords and the high plateau of central Iceland.

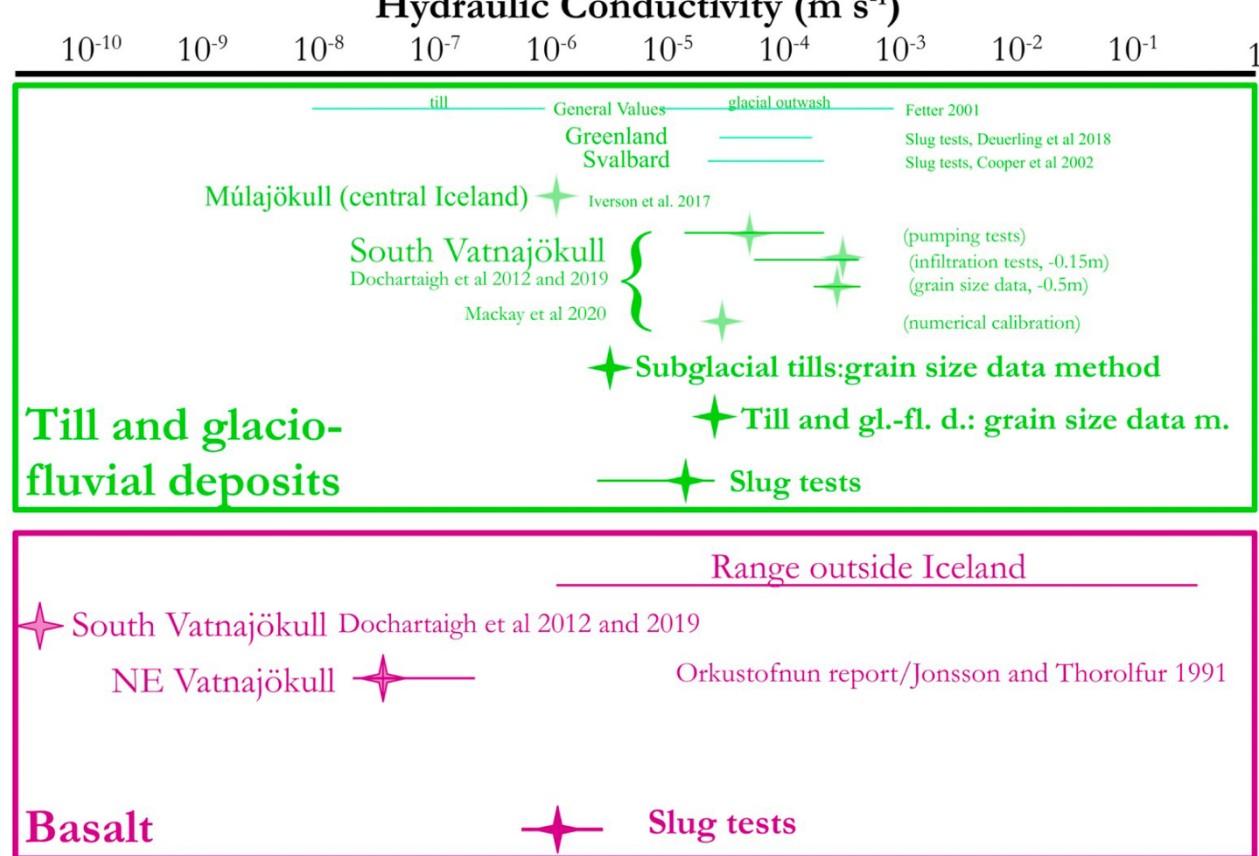

**Figure 11: Hydraulic conductivity values for the till and glacio-fluvial deposits and the basalt from data analysis and the literature; stars indicate single or average value and horizontal lines the range of values. The range for hydraulic conductivities of basalt outside Iceland is based on several bibliographical references, details and references in the text.**

**5.2.3. Aquifers storage coefficients**

Aquifer storage coefficient $S$ is composed of two parts, specific yield $S_y$ and specific storage $S_s$: $S_y + S_s*e$ ($e$ being the thickness of the aquifer), $S_y$ dominates in unconfined aquifers, and $S_s*e$ dominates in confined aquifers. We calculated $S_y$ using the WTF method for three to seven distinct rainfall-recharge events depending on the borehole and the length of the available groundwater record (table 7). Adequate data was available from four boreholes in the till and glacio-fluvial deposits (FLA1, FLA2, FLA3, and FLA4), and two in the basalt aquifer (HA13 and ASK102). Direct use of the WTF method from different values of $S_y$ show that values greater or equal to 0.12 for the till and glacio-fluvial deposits and 0.11 for the basalt imply a recharge larger than the measured precipitation, thus setting an upper limit to the possible interval for $S_y$. The conclusion is that till and glacio-fluvial deposits $S_y \in [0.01–0.12]$ and basalt $S_y \in [0.02–0.11]$.

| | from Grain size data | | From WTF method | From the literature |
|---|---|---|---|---|
| **Geological formation** | **interval $d_{90}$ (µm)** | **$S_y$ (-)** | **$S_y$ (-)** | **$S_y$ (-)** |
| Proglacial till and gl.-fl. d. | 120–760 | 0.12–0.32 | 0.05 (+/- 0.05) | 0.06–0.16 (Morris and Johnson, 1967) |
| Basalt | _ | _ | 0.03 (+/- 0.01) | 0.08 (Heath, 1983) |

**Table 7: Specific yield ($S_y$) values calculated using: (i) An estimate from grain size data (for the till and glacio-fluvial deposits only); and (ii) The Water Table Fluctuation (WTF) method on up to 7 rainfall-recharge events in October 2021 and April and May 2022, average (+/- standard deviation); gl.-fl. d.: glacio-fluvial deposits.**

## 5.3. Aquifers dynamics characteristics

In the studied area, groundwater flows towards the sea. Hydraulic gradients are deduced from the difference in groundwater levels between two boreholes on the same potential groundwater flow line. The estimated hydraulic gradient in the till and glacio-fluvial deposits aquifer is approximately 4.5/1000 south of Fláajökull. The hydraulic gradient in the basalt aquifer is approximately 3.5/1000 south of Fláajökull, 3.9/1000 south of Skálafellsjökull and 30/1000 in Hali.

Time evolution of the groundwater level in the till and glacio-fluvial deposits aquifer (figures 7 and 8) show clear recharge events by rainfall, snow melt, and glacial meltwater. Recharge by rainfall events occurs within 24 hours of the precipitation event, at least when the rain is > 10 mm. Snow melt events are identified in February-March 2022 , and also on a shorter time scale in January 2022. When the precipitation is snowfall and the snow cover lasts for more than one day, the lag between a snowfall precipitation event and the recharge of the groundwater is visible (fig. 7 and 8), corresponding to the time that it takes for the snow to melt. In September the larger increase of the water level in borehole FLA3 (+0.54 m from 30 August 2022 to 21 September 2022) compared to FLA2 and FLA1 (+0.21 m and +0.11 m, respectively, during the same period; fig. 7) demonstrates additional recharge by glacial meltwater. Similarly in the basalt aquifer (fig. 8) the trend line of the water level in borehole ASK105 shows an increase of +0.40 m from 17 August 2022 to 19 September 2022 that cannot be accounted for only by the precipitation events during the same period. ASK105 is 4.2 km from the nearest glacier terminus; boreholes further away show a similar but smaller increase during the same period of time: ASK103 6.2 km from terminus: +0.2 m; ASK102 at 7.4 km from terminus: +0.22 m; HA16 at 7.4 km from terminus: +0.19 m.

Temperature data from FLA4 borehole exhibit 4 plateaus (constant value over a period of time) of temperatures between 4.2 to 5.4 °C (fig. 12). These plateaus correspond to every time the water level is lower than 24.1 m a.s.l. (fig. 12). We interpret this plateau as an upward leakage from the basalt aquifer, triggered when the water level in the till and glacio-fluvial deposits aquifer is lower than the piezometric level in the confined basalt aquifer. The fact that these plateaus occur at different periods of the year (September, December, January and February) demonstrate that groundwater level in the basalt aquifer must therefore be relatively constant over the period. The temperature measured in the basalt aquifer in ASK105 (1.6 km from FLA4), from 5 to 9 °C, corroborates that hypothesis.

The clear separation of both aquifers, and the confined characteristics of the basalt aquifer observed in some locations, lead us to hypothesize the presence of a much less permeable layer between the till and glacio-fluvial deposits and the basalt formation. It could be a clay layer or a more compacted till.

At the coastline, 3 hypotheses can be made for the fresh/marine groundwater interface for each aquifer: equilibrium around the coastline, fresh groundwater pushing the interface offshore, or marine intrusion inland. The few data acquired on the EC of the groundwater (high EC values near the coastline, table 2) suggest a potential seawater intrusion in the basalt aquifer. For the till and glacio-fluvial aquifer we expect that some fresh groundwater is flowing into the sea as this aquifer is unconfined and in direct connection with the sea.

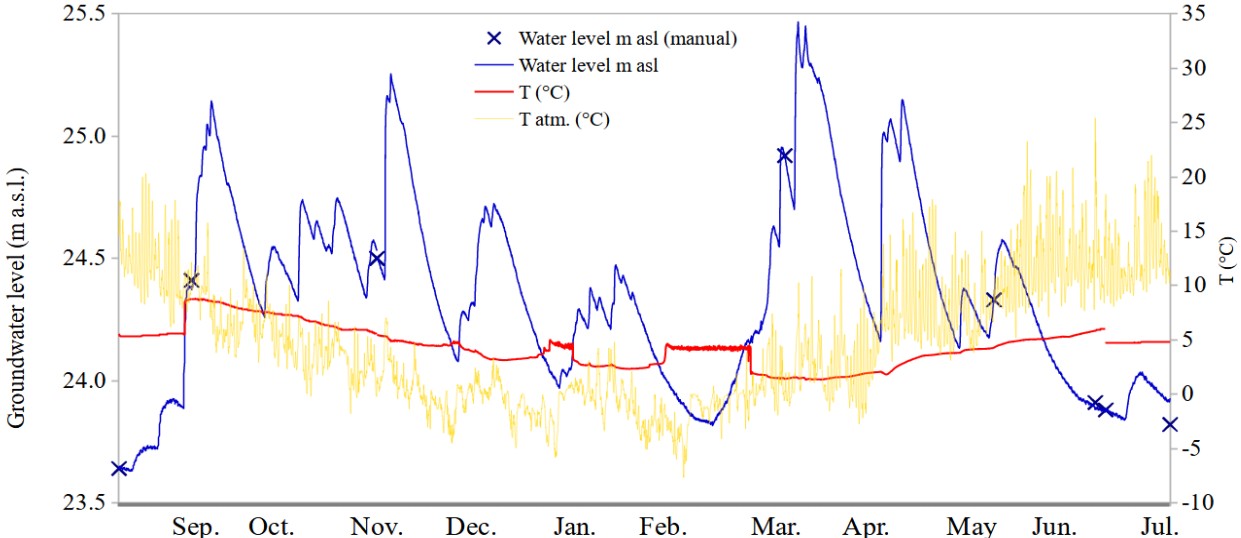

**Figure 12: Hourly groundwater level (in blue, m a.s.l.) and temperature (in red) in the borehole FLA4, and atmospheric temperature (in yellow, from the Baro Diver probe in borehole FLA4). The temperature chronicle exhibits plateaus around 5 °C at the end of August 2021, in January, February, and July 2022, each time the groundwater level is lower than 24.1 m a.s.l.**

## 6. Hydrogeological conceptual model of glacierized catchments

Step by step, the compilation of available data and the collection of new data is leading to a better understanding of the role of groundwater in the water cycle of glacierized catchments. It is now possible to quantify the complete water balance and the amount of subglacial and proglacial recharge to the aquifers. To validate this latter, a first-order-of-magnitude sensitivity analysis was performed with a hydrogeological numerical model (MODFLOW Model Muse® (Harbaugh et al., 2000; Winston, 2019; Winston et al., 2017). The model was built using the geometry, and the hydrodynamic parameters obtained in this study. Several geometry configurations were tested: 1 layer (till and fluvio-glacial deposits), 2 layers (till and fluvio-glacial deposit, and the basalt), and 3 layers where an aquitard between the two aquifers were interposed. For each configuration, the tests were done to check the values of the hydrodynamic boundary conditions (i.e.: The recharge to the aquifer) able to reproduce the range of the water level observed in the aquifers of the proglacial area. The results show that the configuration including an aquitard offers the best promising results and part of the subglacial flow has to contribute to the groundwater flow.

In the subglacial area, the subglacial recharge to the aquifers can represent 50% of the glacial melt, with a range of 50% (i.e.: 25-75% of the glacial melt water). This ratio is in the order of magnitude of the one deduced from a 2012-2013 surface runoff data set (Young et a., 2015). In this study, a 67.3 km$^2$ catchment of the glacier is studied, with daily surface runoff measurements in a 500 m downstream river south of Skálafellsjökull. The surface runoff measured and extrapolated (for days without data), that they obtained, accounts for 50 % (± 10%) of the volume of ice melt during the same period.

Between the glacier terminus and the coastline (i.e.: In the proglacial area), to quantify the recharge to the aquifers we have now the maximum possible amount (i.e.: The effective rainfall). In first, approximation, the scaling coefficient varying of 0.5 (with a range of 0.25–0.75) is applied to the effective rainfall. That is based on field observations (important surface runoff), and in the absence of surface flow record allowing a more accurate estimation (table 8). The interannual average hydrologic balance on 1990–2021 on the proglacial area can thus be detailed as the following: 1540

mm (± 310 mm) of precipitation, 520 mm (± 80 mm) of evapotranspiration (both methods used), 500 mm (± 150 mm) of runoff and 500 mm (± 150 mm) of recharge to the aquifers.

The subglacial recharge rate is estimated based on available data to be on average 2000 mm year$^{-1}$ and in the proglacial area 500 mm year$^{-1}$ (table 8). Thus, our estimates indicate that the subglacial recharge is about 4 times higher than the one on the proglacial area, which is consistent with studies claiming a high recharge of the till and glacio-fluvial deposits aquifer by the melting of the glaciers (Sigurðsson, 1990; Xiang et al., 2016), while offering additional quantitative comparison. Following the patterns of the available water for both subglacial runoff and groundwater recharge (see fig. 10 and table 5), the subglacial recharge is highly seasonally variable (fig. 10) and is much higher under the lowest elevation of the glaciers (table 5).

| Scaling coefficient applied | Subglacial recharge (mm year$^{-1}$) | | | |
|---|---|---|---|---|
| | From Summer Mass Balance | | | From HH-ICRA results |
| | 2010-2021 | 2010-2016 | 2021 | 2010-2016 |
| 25% | 1000 | 1000 | 1000 | 1100 |
| **50%** | **1900** | **1900** | **2000** | **2100** |
| 75% | 2900 | 2900 | 3100 | 3200 |
| | Recharge in the proglacial area between the glacier's terminus and the coastline (mm year$^{-1}$) | | | | |
| | From PET calc. with Thornthwaite method | | | From PET calc. with Penman-Montheith method | | |
| | 1990-2019 | 2020 | 2021 | 1990-2019 | 2020 | 2021 |
| 25% | 260 | 280 | 220 | 360 | 220 | 210 |
| **50%** | **510** | **560** | **450** | **480** | **450** | **410** |
| 75% | 760 | 840 | 670 | 730 | 670 | 620 |

**Table 8: Recharge to the aquifers (mm year$^{-1}$) in the proglacial area (between the glaciers terminus and the coastline), estimated from the effective rainfall, and in the subglacial area, estimated from i) from summer mass balances (field data, IES Glaciology group) and effective rainfall, ii) from HH-ICRA results (Schmidt et al., 2020) and effective rainfall. calc.: calculated.**

The conceptual model of the groundwater dynamics characteristics in glacierized catchments derived from all the analyzed data is presented in figure 13. A glacier with crevasses and moulins that provide pathways for the surface meltwater to the subglacial water system is presented, along with the two geological formations underlying it and in its front: The till and glacio-fluvial deposits and the basalt formation. The average characteristics of the hydrogeological formations are described along with their spatial variability (thickness, hydraulic conductivity, specific yield) and/or seasonal variability (temperature, electrical-conductivity). The subglacial recharge and recharge in the proglacial area are quantified and their monthly variability shown in the bottom figures. The groundwater flow is indicated with arrows. The water is entering the basin as precipitations, immediately joining the hydrologic and hydrogeological system if rainfall, or joining it delayed through melted snow and melted ice. Both the surface and groundwater flows are towards the coast. Exchanges between the subglacial hydrology network and the till and glacio-fluvial aquifer occur, as well as between the surface hydrology network and the till and glacio-fluvial aquifer. Downward (recharge) and also upward (leakage) exchanges occur between the tills and glacio-fluvial aquifer and the basalt aquifer.

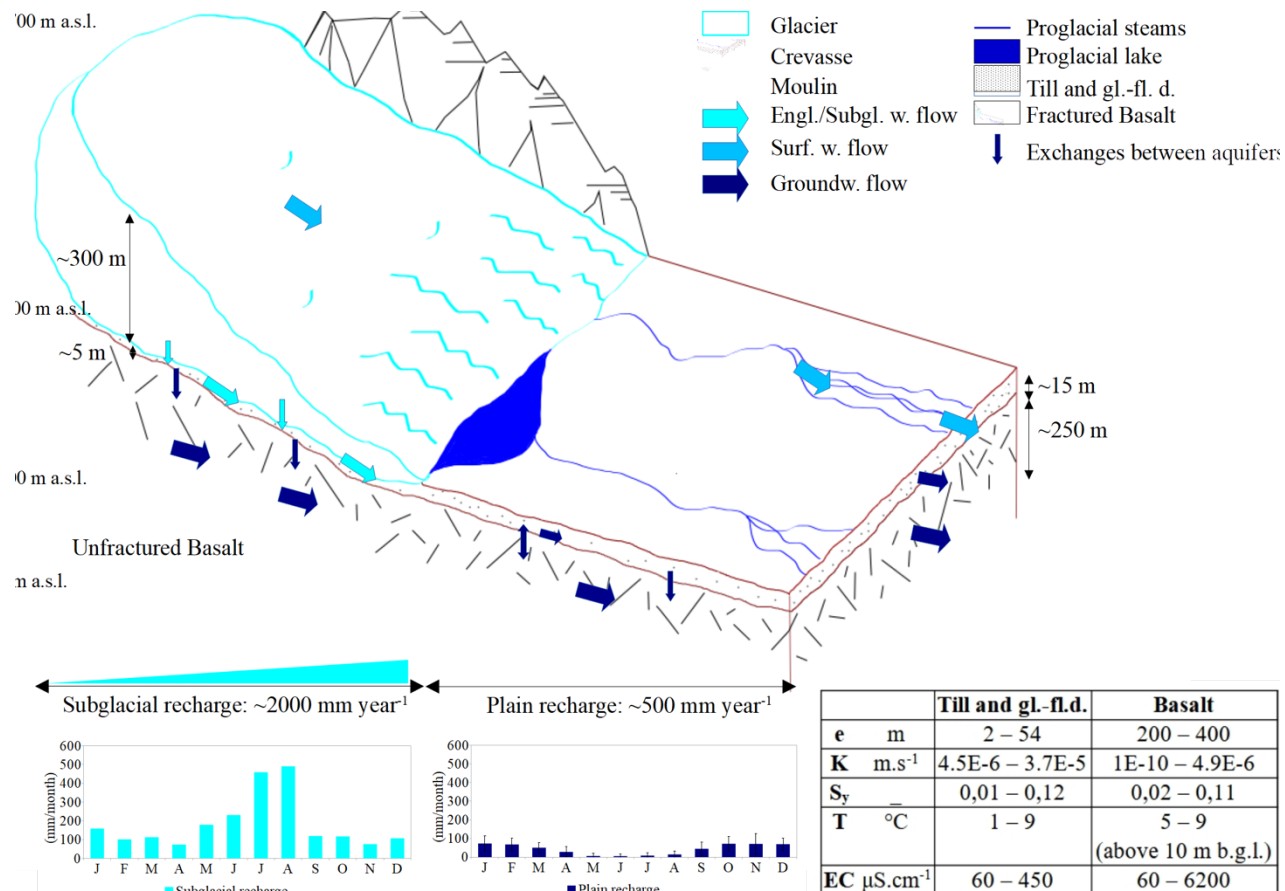

Glacier
Crevasse
Moulin
Engl./Subgl. w. flow
Surf. w. flow
Groundw. flow

Proglacial steams
Proglacial lake
Till and gl.-fl. d.
Fractured Basalt
Exchanges between aquifers

~300 m

~5 m

~15 m

~250 m

Unfractured Basalt

Subglacial recharge: ~2000 mm year⁻¹

Plain recharge: ~500 mm year⁻¹

Subglacial recharge

Plain recharge

| | | Till and gl.-fl.d. | Basalt |
|---|---|---|---|
| e | m | 2 – 54 | 200 – 400 |
| K | m.s⁻¹ | 4.5E-6 – 3.7E-5 | 1E-10 – 4.9E-6 |
| $S_y$ | – | 0,01 – 0,12 | 0,02 – 0,11 |
| T | °C | 1 – 9 | 5 – 9 (above 10 m b.g.l.) |
| EC | µS.cm⁻¹ | 60 – 450 | 60 – 6200 |

**Figure 13: Conceptual model of the groundwater dynamic in glacierized catchments such as those observed at Fláajökull, Heinabergsjökull and Skálafellsjökull. The model is based on all the data presented in this paper. A glacier is at the head of the catchment along with the two geological formations underlying it and in its front: The till and glacio-fluvial deposits and the basalt formation. Average characteristics of the hydrogeological formations are described along with their spatial (e, K, $S_y$) and seasonal (T, EC) variability as well as recharge rates (subglacial and on the proglacial area) along with their monthly variability. The groundwater dynamic is expressed by arrows. Subgl.: subglacial; w.: water; surf.: surface; groundw.: groundwater; gl.fl. d.: glacio-fluvial deposits.**

## 7. Conclusion

Thanks to the new data sets presented in this paper, acquired notably about the groundwater compartment, and their analysis it was possible to design a new conceptual model of glacio-hydrogeological behavior in glacierized catchments.

We have identified two different aquifers, one in the till and glacio-fluvial deposits and one in the basalts, with different hydrodynamic behaviors, using the geological map, and then measurements in the boreholes of the new observation network established. For the till and glacio-fluvial deposits aquifer, we estimated a hydraulic gradient of 4 to 5/1000, and from 3 to 30/1000 for the basalt aquifer. By the borehole FLA4 closest to the glacier's terminus, when the water level in the till and glacio-fluvial deposits aquifer is lower than the piezometric level in the basalt aquifer, a vertical upward leakage from the basalt aquifer takes place.

We have calculated the hydraulic conductivities and specific yields of both aquifers from field measurements: i) Till and glacio-fluvial deposits: $K \in$ [4.5E-6–3.7E-5] m s⁻¹, $S_y \in$ [0.01–0.12]; ii) Basalt aquifer: $K \in$ [1E-10–4.9E-6] m s⁻¹, $S_y \in$ [0.02–0.11]. These compare well with the values extracted from the scientific literature (fig. 11 and table 7). For the $K$

of the basalt aquifer, it is a much narrower range than in the literature, however, the values we have gathered are not sufficient to determine the spatial variability.

We have obtained a comprehensive water balance for the whole glacierized catchment, from the estimation of the water available for surface runoff and recharge of the aquifer system both under the glacier and in the proglacial area. Recharge under the glacier is estimated to be about 4 times higher than the one on the proglacial area, which is consistent with studies claiming a high recharge of the till and glacio-fluvial deposits aquifer by the melting of the glaciers. The subglacial recharge has a large seasonal variability (for 2021 maximum in July and August about 480 mm

560 month$^{-1}$, and minimum in April and November, about 80 mm month$^{-1}$) and with the elevation (highest at the lowest glacier elevation 3400 mm year$^{-1}$ in 2010, lowest at the highest elevation: less than 1000 mm year$^{-1}$ in 2010).

Several unknowns remain: 1) The saturation of the till and glacio-fluvial deposits and basalt formations under the glacier. 2) The presence of a clay layer or compacted till between the subglacial till and the basalt on the subglacial area, and also between the till and glacio-fluvial deposits and the basalt formation on the proglacial area, is to be confirmed.

It would explain the separation of the two aquifers and the confined character (even artesian one) of the basalt aquifer observed in some locations. 3) The connection of the discharge in the aquifers with the ocean, we expect that some fresh groundwater from the till and glacio-fluvial deposits is flowing into the ocean, for which a hydrogeological numerical model is required to calculate an estimate of the groundwater flux to the sea. A model would also help to test the hypothesis of a slight seawater intrusion in the basalt aquifer.

We are thus developing numerical hydrogeological models that will allow further assessments of the unknowns. Manual and automatic groundwater level measurements are to be continued for several years. Eventually, to further explore the potential feedback of the groundwater system on the glacier dynamics, a coupling of a glaciological model and the hydrogeological model would be necessary.

As demonstrated, subglacial recharge to aquifers is significant for warm-based glaciers lying on sediments and/or

575 fractured bedrock. It is thus important to include the groundwater component to evaluate accurately the water balance of glacierized catchments.

**Data availability** All new data acquired for this paper and described in section 4 are openly available, without any restrictions and along with the associated metadata on Zenodo repository: SlugTests Data:

https://zenodo.org/record/7716507; IceAq Groundwater Hourly Data: https://zenodo.org/record/7716453; IceAq Groundwater Monthly Manual Data: https://zenodo.org/record/7716362.

**Authors contribution** AV devised the project. AV planned the field work and carried it out along with CD, OF, SV, GA and SG. JH provided data from field work carried out in 2012-2013. FP provided summer mass balance data. AV, CD

and OF treated the data. AV, SV, GA, CD and OF contributed to the interpretation of the results. AV wrote the manuscript with support from SV and critical feedbacks from all authors.

**Funding** IceAq project has received funding from the European Union's Horizon 2020 research and innovation program under the Marie Skłodowska-Curie grant agreement n° 951732. Extra support was received from Campus

France PHC Jules Verne funding (French-Icelandic partnership, agreement n°48330SL) and two Erasmus+ grants.

**Acknowledgements** Michael Pettersson for field work assistance from May 2021 to September 2022, Haukur Ingi Einarsson from Glacier Adventure in Hali for his help with the groundwater level data acquisition, Maud Bernat for field assistance in June 2022, and several farmers for access to their land. The authors acknowledge Christophe Cudennec, Xuegao Chen, and anonymous reviewers for their constructive comments on previous version of the manuscript.

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
