# Peer review of "A hydrogeological conceptual model of aquifers in catchments headed by temperate glaciers"

_EGUsphere, 2022_

## Author Response (AR1)

*Christophe Cudennec Review:*

*The article presents a highly valuable setup for new observations, a comprehensive presentation of pre-existing / newly acquired / proxy data on all involved compartments and processes, in an area which is one of the most emblematic ones in terms of glacier retreat - yet much unknown and ungauged so far. The articles elaborates on the new data acquired and related ones to propose first quantification and a conceptual model.*

*I think the observation/data rationale and the initial database presented, as well as the conceptual model are very relevant and timely results. Quantifications presented in the paper are an interesting initial assessment which will definitely be refined and deepened in the next steps thanks to additional months of data, adjustments/improvements allowed by the loop back from the conceptual model to the field and data reanalysis, and eventual modellings. The full transparency about the data is a strong quality of this paper, which paves the way to precise next steps and will thus become a seminal paper for anyone in the future using the full original open database. Yet, actual uncertainties and future steps about quantification should be made more explicit here, so that the paper is clearer about the strong achievements, the first quantifications proposed, and the future steps.*

**Uncertainties have been added systematically to all values measured or calculated when they were not explicit.**

*The conceptual model in Figure 13 could be detailed a bit further to make the scheme more complete. In particular the elevation above sea level could be added and the fresh/marine groundwater interfacing could be addressed/questionned.*

**The figure 13 has been modified, it now includes the elevation above sea level. The fresh/marine groundwater interface is discussed in further detials in the discussion section.**

*I wonder about the quantification of some meteorological terms: 1) is'nt there any estimate of the snow depth over the ice cap / glaciers to complement the met stations located in the Hofn neighbourhood, which is close to sea level and so leads to issues about representativeness?*

**To our knowledge there is no estimate of the snow depth over the glaciers; but the snow cover (total/partial/inexistant) observed from Höfn is provided for the plain and for the mountain.**

*2) how are evapotranspiration methods parameterized in such a particular, lowly vegetated environment?*

**About the evapotranspiration methods: we used Thornthwaite and Penman Monteith methods, with parameters adapted to the latitude. We chose classic methods, as from the literature there is no specific method proved more representative for Iceland. It has been made explicit in the text of the paper. For the calculation of the effective rainfall, the parameter describing the soil is adapted. It should be noted too that about half the plain is cultivated, on former wetland providing thick soils.**

*Looking forward to see the revised version, C. Cudennec*

*Anonymous Review:*

**Global corrections done:**

**- Introduction: clarification that the objectives of the paper are the understanding of the whole hydrogeological system as a whole, including the recharge, as well as the geometry and hydraulic parameters of the aquifers;**

**- Methodology and results: we are now focusing first on the available water for surface flow and groundwater flow (effective rainfall in the plain, total of subglacial melt and effective rainfall under the glacier), and discuss its contribution to recharge afterwards, in the Discussion section;**

**- The conceptual model presented in section 5.3 is described and discussed more in details.**

*The authors present a study aiming at achieving the quantification of the groundwater recharge under the glaciers and in the plain and proposing a hydrogeological conceptual model. Overall, the topic is interesting and the authors provide new data about groundwater dynamics. However, the paper is more of a descriptive write up for the observation data. The manuscript is analyzing the characteristics of the groundwater level, temperature, EC, and aquifer information, and giving rough estimation results of groundwater recharge. Two parts of new data analysis and recharge estimation are relatively independent. The paper doesn't have enough novelty and it doesn't satisfy the required standards for the journal*

*1. I have some doubts on the validity of the calculations in section 5.1 and 5.2. Half of the glacier melt water is considered to infiltrate to the aquifers under the glaciers. A scaling coefficient of 0.5 is applied to the effective rainfall to obtain the recharge rate in the plain. The ranges of the surface runoff in the glacier hillslope and plain area of four glacial catchments are wide in different seasons. The authors are encouraged to carry out a simple quantitative water balance analysis (both in the glacier and the plain area) to assess the uncertainties in the data and the reliability of conclusions they make regarding the negligible impact of the change in the surface runoff etc*

**1. See global corrections underway mentioned above. In the current version of the paper see ll.176-181 and ll.391-397.**

*2. Even if the calculations are accepted, I think that the authors do not give additional discussion on the comparison with the previous studies, from which the readers can already obtain the same conclusion: recharge under the glacier is much higher than the one on the plain. The authors should give more discussions on the new findings in this paper and the comparisons with previous researches.*

**2. A Discussion section has been created, including such discussion.**

*3. The manuscript doesn't address how the complex and dynamic relationships between the groundwater recharge and geological conditions. The newly measured geological conditions and groundwater dynamic processes in this study are interesting but without a more targeted purpose for this paper.*

*The water exchange between two aquifers (drawn in Figure 13) is not mentioned and discussed in the manuscript.*

**3. See global corrections mentioned above. The upward leakage is discussed in the current version of the paper ll. 398-403.**

*4. The paper is limited to the presentation and simple analysis of various collected data. The reviewer feels that the title "A hydrogeological conceptual model of aquifers in catchments headed by temperate glaciers" does not reflect the content of the paper.*

**4. See global corrections mentioned above.**

*Here, some detailed comments are listed:*

*1. Chapter 1-introduction: What's the current study progress on the research objectives? including the groundwater recharge in glacial catchments and the hydrogeological conceptual model. In addition to the lack of data, what are the deficiencies in the study of the hydrogeological conceptual model, and what is the new development of this research?*

**1. The introduction has been detailed on those aspects.**

*2. Regarding the first goal of "proving whether or not meltwater from the glacier recharges the aquifer(s) beneath the icesheet", there is very little reflection on this. The authors assumed that half of the melt infiltrated to the aquifers (P.9, L.175). Proof is given ll.391-397 + J.Hart data, from which the 50% quotient is deduced (ll.176-181 ). revoir redaction methodo: we need estimation of subglacial melt and of % of melt infiltrating; appuyer sur difference echelle et plus incertitude*

**2. See global corrections mentioned above and ll.391-397.**

*3. Section 3.2 Data analysis: "if Pi + Si < PETi , RETi = max(Pi + Si ; PETi )"(P.8, L.156-158), is this correct? The authors have to carefully check these equations that are used to calculate the effective rainfall.*

**3. Equations checked.**

*4. The contents of section 4-New data and section 5.2-Aquifers characteristics can be merged together to show results of monitored data.*

**4. Indeed, they were separated to avoid a too long section in 5.2 by presenting the raw data earlier.**

*5. The authors spend a lot of time introducing the aquifer thickness, hydraulic conductivity, storage coefficient, and electro-conductivity. However, all these monitored data have nothing to do with the research objective of estimating the recharge process. What is the impact of geological conditions on the recharge?*

**5. See global corrections mentioned above.**

*6. It is wrong for the format of "Gardner et al 2013"(P.2, L.28), "Einarsson 1994"(P.6, L.97), "Torfason 1979" (P.6, L.100), "Jóhannesson and Sæmundsson 1998" (P.6, L.103), etc.*

**6. References format have been corrected when necessary.**

*7. P.8, L.159: "EvapoTranspiration"should be corrected.*

**7. Typo correction done.**

*8. P.8, L.160: "SW is initially estimated at 50mm", what is the mean of SW?*

**8. SW is the same as S, erroneous double notations, corrected.**

*9. Figure 7: the colors are not clear to distinguish the results of different boreholes. The "bgl"in the figure should be corrected as "b.g.l.".*

**9. Typo correction done.**

*10. P.14, L.283: "Subglacial recharge, estimated from SNMELT variable", what is the mean of SNMELT?*

**10. SNMELT is the melt in the PISM-ICRA model, sentence clarified.**

*11. P.15, L.292: "The monthly variation of the effective rainfall, based on the Potential Evapotranspiration (PET)", however, the Potential Evapotranspirationfirst appears in L.152.*

**11. Evapotranspiration (PET) now written l.152.**

*12. P.21, L.433-435: "Recharge under the glacier is 4 times higher than the one onthe plain, which is consistent with studies claiming a high recharge of the till and glacio-fluvial deposits aquifer by themelting of the glaciers (Sigurðsson, 1990; Xiang et al., 2016)." The references should be removed in the section of Conclusion.*

**12. References moved from the conclusion to the discussion section.**

---

## Author Response (AR2)

Response to the reviewer's comment

We thank the editor and reviewers for reviewing our manuscript and considering our work. We appreciate their interest in our research and their constructive comments have enabled us to improve the manuscript, to better highlight the main results and to strengthen the demonstration that has led us to develop a new conceptual model of the hydrogeological functioning of glacierized catchments.

Reviewer Report of Xuegao Chen, reviewer1:

The authors have done a good job implementing comments from the first round of revisions which has improved the manuscript. Overall, the general idea of the work is interesting and the paper is streamlined. The main strong points of the paper are two. First, it provides new data about aquifer systems in four south-eastern outlet glaciers. The monitored groundwater level, aquifer temperature, EC and calculated hydraulic conductivity and specific yield give an overall picture of the local hydrogeologic condition. Second, it gives quantitative information on the recharge process from the glacier to the aquifer based on the watershed water balance. The glacial melt recharge has a significant impact on the groundwater dynamic. I think the manuscript can be recommended for publishing and I wish them success in their research.

We are grateful to Dr. Xuegao Chen for his appreciation of our work.

Anonymous reviewer 2
This manuscript tries to understand and characterize the hydrogeological system including the recharge, geometry, and hydraulic parameters in a glacier feed region. The main contents are based on new observations, and a hydrogeological conceptual model of the system is proposed by analyzing the existing and new data. However, I am confused at the current structure of the manuscript and there are no interconnections among different part. As a result, I could hardly get what the authors want to emphasize. The current version seems like a part of a report. I recommend a rejection at current stage and my comments are listed as below:
(1) The title is "a hydrogeological conceptual model of aquifers ......", However, this model is finally proposed in the discussion part with the initial introduction of the concept framework. The structure, validation, and the performance are missing the manuscript.

The structure of the manuscript has been re-worked and improved, taking into account this major comment. Paragraphs linking the sub-sections provide a clearer understanding of the step-by-step reasoning behind the construction of the conceptual hydrogeological model.

(2) The research gap, motivation are not clearly stated in the introduction part. I could hardly get the main point that the authors want to deliver to the readers.

The introduction has been rewritten to better present the state of the art in this particular context, the scientific questions we wish to answer, the main advantages of the study site, and the type of new data that has been collected and brought together with that available to be able to propose a conceptual hydrogeological model of the glacierized catchments.

(3) The authors need to provide a detailed explanation on how the "new data" assists the analyses.

Until now, the role of the underground compartment in glacial watersheds has been little studied. Very few studies provide data on the geometry of aquifers, their hydrodynamic properties, or their piezometric records. This is even less the case if a confined aquifer, deeper than the detrital aquifer, is to be considered. The originality of our study lies in our ability to collect new data on the aquifer systems that make up the site studied. We can thus describe the geometry of the system. Analysis of the piezometric records shows not only recharge by glacier meltwater, but also vertical exchanges between the two aquifers, via an aquitard. These data, combined with those on subglacial melt flows and estimates of recharge in proglacial zones, enable us to draw up a complete hydrological balance for this type of environment. The rewriting and reworked structure of the manuscript should make it easier to understand the stages of our argument and how our new data contributes to it.

(4) Most of the subtitles need to be revised, for example. 5.1.2 "in the plain", 4 "new data" (5) The authors need to re-organize the manuscript and clearly point out the main research question that needs to be addressed.

As mentioned in point 1), the structure of the manuscript has been reworked, including the subtitles. The plan is now as follows (modified or added subtitles are underlined in yellow):

**A hydrogeological conceptual model of aquifers in catchments headed by temperate glaciers**
**1. Introduction**
**2. The study area**
2.1. Climate context
2.2. Glaciers context
2.3. Geological context
**3 Methodology**
3.1 New Data
3.1.1. Aquifers geometry
3.1.1. Aquifers dynamics and properties
3.2 Data Analysis
3.2.1. Glacier melt and effective rainfall
3.2.2. Hydrodynamic properties of aquifers
**4. Hydrodynamic characterizations of the aquifers**
4.1. Groundwater level
4.2 Temperature
4.3 Electro-conductivity
4.4 Slug tests
**5. Results**
5.1 Recharge estimation
5.1.1 Estimation of subglacial water flows and spatial distribution
5.1.2 Estimated effective rainfall in the proglacial area
5.2 Aquifers characteristics
5.2.1 Thickness and temperature of geological formations
5.2.2 Aquifers hydraulic conductivity
5.2.3 Aquifers storage coefficients
5.3 Aquifers dynamics characteristics
**6. Hydrogeological conceptual model of glacierized catchments**
**7 Conclusion**

---

## Author Response (AR3)

Response to the reviewer's comment

We would like to thank the editor and the reviewer for reviewing our manuscript and taking our work into account. In this revised version, we have responded point-by-point to the three main comments made by the sole reviewer. We have thus made all the necessary changes to the text requested by the reviewer and offer a clearer version of our manuscript. We hope that both the reviewer and the editor will agree with us so that the publication of our paper is not unnecessarily delayed.

Please find below the details of our point-by-point responses to the reviewer.

Reviewer Comment - Major Revision:
"The authors put great efforts to response to the comments, however, some critical issues still exist and need to be addressed before the manuscript could be accepted.
(1) based on the revised main text, the current title is not appropriate. The main contribution is a new comprehensive understanding of aquifer systems in catchments headed by temperate glaciers based on new collected data. I could hardly get the information on how the model was developed, calibrated, and validated and how the proposed model performs."

**Response to the reviewer**: Since the initial evaluation of our manuscript, the reviewer has disagreed with our title. However, our objective is to present the hydrogeological functioning of a glacierised catchment based on existing and newly collected data. In the scientific literature, it is commonly referred to as a "conceptual hydrogeological model". For example, this technical term was used for a previous paper published in HESS, entitled "Hydrogeological conceptual model of andesitic watersheds revealed by high-resolution heliborne geophysics" (Vittecoq et al., HEES 2019, https://doi.org/10.5194/hess-23-2321-2019). We respectfully disagree with the reviewer's comment and wish to retain the title as it accurately reflects our objective.
In the discussion of the previous version submitted in February 2024, we included some information on a numerical modeling project in progress. We recognize that this may have caused confusion regarding the use of the technical term "model". Therefore, this paragraph has been removed in the new version, as its added value is minimal.

(2) Is Section 4 part of the results? Section 4.4 is more like a method; it should be something like "Hydraulic conductivities." The authors should take more efforts to re-organize the structure of the manuscript.

**Response to the reviewer**: We concur that section 4 is part of the results based on new data acquisition. To avoid any confusion, the structure of the manuscript has been reorganized. The subtitle "4.4 - slug-tests" has been replaced by "Hydraulic conductivities" in subsection 4.1.2. Please find below the new plan for the manuscript. The changes are highlighted in yellow for those made in response to the "n-1" comments and in green for those made in response to the "n" comments:
**A hydrogeological conceptual model of aquifers in catchments headed by temperate glaciers**
**1. Introduction**
**2. The study area**
2.1. Climate context
2.2. Glaciers context
2.3. Geological context
**3 Methodology**
3.1 New Data
3.1.1. Aquifers geometry

(3) The authors should provide the main revised part directly under each comment, currently, the revised part is highlighted without a mark (response to comment X), I could hardly get the how the authors revised to each individual comment.

**Response to the reviewer**: In the previous resubmission (February 2024), a highlighted version was uploaded to the EGUSPHERE server. Changes to the manuscript were highlighted in yellow. The current highlighted version shows the changes made in response to the "n-1" comments in yellow and the ones to the "n" comments in. Marks have also been added to indicate which comment each change responds to. As the manuscript has been reorganized, it may indeed be difficult to identify all the changes made. However, at the same time, the current and previous comments remain rather vague. We did our best to use them in a constructive way.